# On Expressive Power of Looped Transformers: Theoretical Analysis and Enhancement via Timestep Encoding

## Abstract

Looped Transformers offer advantages in parameter efficiency and Turing completeness. However, their expressive power for function approximation and approximation rate remains underexplored. In this paper, we establish approximation rates of Looped Transformers by defining the concept of the modulus of continuity for sequence-to-sequence functions. This reveals a limitation specific to the looped architecture. That is, the analysis prompts us to incorporate scaling parameters for each loop, conditioned on timestep encoding. Experimental results demonstrate that increasing the number of loops enhances performance, with further gains achieved through the timestep encoding architecture.

## 1 Introduction

Transformers (Vaswani et al., 2017) have become the standard neural network architecture for a wide range of machine learning tasks, such as natural language processing and computer vision. Looped Transformers have an architecture composed of fixed-size Transformer layers, where the output is fed back into the input. This structure offers advantages over standard Transformers, such as inductive bias, parameter efficiency, and Turing completeness. Dehghani et al. (2019) first introduced the idea of incorporating recursive inductive bias into Transformers, aiming to address tasks that RNNs handle more easily. Looped Transformers are also related to weight-tying Transformers (Lan et al., 2020), demonstrating comparable performance to standard Transformers with fewer parameters. More recently, Giannou et al. (2023) theoretically demonstrated that the recursive structure of Looped Transformers allows them to function as Turing machines. In summary, Looped Transformers are more parameter-efficient and potentially more expressive than standard Transformers, enhancing their generalization capabilities.

The expressive power of standard Transformers has been extensively explored, showing that they can approximate continuous permutation-equivariant functions on compact domains (Yun et al., 2020; Kim et al., 2023; Takakura & Suzuki, 2023; Jiang & Li, 2024; Kajitsuka & Sato, 2024). In addition, their approximation rate has been studied: Takakura & Suzuki (2023); Jiang & Li (2024) established it by identifying the properties of the target functions, revealing the types of functions that Transformers can approximate effectively. In contrast, the expressive power of Looped Transformers in function approximation remains unexplored. Due to the structural constraints imposed by weight-tying, which limit their flexibility, existing universal approximation theories for Transformers cannot be directly applied. Moreover, the approximation rate and the appropriate properties of the target functions for Looped Transformers have yet to be investigated. Recently, Zhang et al. (2023) first explored the expressive power of *looped* models. They established an approximation rate for ReLU networks for continuous functions in terms of loop counts and modulus of continuity. Still, this remains unclear in the case of Looped Transformers.

In this paper, we derive the approximation rate of Looped Transformers for continuous sequence-to-sequence functions by defining the concept of *sequence continuity*, *contextual continuity*, and *token continuity*. This reveals a limitation specific to the looped architecture. That is, the analysis prompts us to incorporate scaling parameters for each loop, conditioned on timestep encoding.

## 2 BACKGROUND

We define the Transformer architecture in Section 2.1 and Looped Transformers in Section 2.2. We then introduce related work of theoretical analysis on function approximation power of Transformers in Section 2.3.

### 2.1 TRANSFORMER ARCHITECTURE

Given an input sequence $\boldsymbol{X} \in \mathbb{R}^{d \times N}$, composed of $N$ token embedding of dimension size $d$, the self-attention layers with $h$ heads and head size $s$, and the feed-forward layer with width size $q$, are defined as follows:

$$\text{Attn}(\boldsymbol{X}) = \sum_{i=1}^{h} \boldsymbol{W}_i^{(O)} \left( \boldsymbol{W}_i^{(V)}(\boldsymbol{X}) \right) \sigma_S \left[ \left( \boldsymbol{W}_i^{(K)}(\boldsymbol{X}) \right)^{\top} \left( \boldsymbol{W}_i^{(Q)}(\boldsymbol{X}) \right) \right] \in \mathbb{R}^{d \times N}, \quad (1)$$

$$\text{FF}(\boldsymbol{X}_{:,i}) = \boldsymbol{W}^2 \cdot \sigma_R(\boldsymbol{W}^{(1)} \cdot \boldsymbol{X}_{:,i} + \boldsymbol{b}^{(1)}) + \boldsymbol{b}^{(2)} \in \mathbb{R}^d, \quad (2)$$

where $\boldsymbol{W}_i^{(V)}, \boldsymbol{W}_i^{(K)}, \boldsymbol{W}_i^{(Q)} \in \mathbb{R}^{s \times d}, \boldsymbol{W}_i^{(O)} \in \mathbb{R}^{d \times s}, \boldsymbol{W}^{(1)} \in \mathbb{R}^{q \times d}, \boldsymbol{W}^{(2)} \in \mathbb{R}^{d \times q}, \boldsymbol{b}^{(1)} \in \mathbb{R}^q, \boldsymbol{b}^{(2)} \in \mathbb{R}^d$ are parameters, $\sigma_R$ denotes ReLU function, and $\sigma_S$ denotes a softmax operator applied to the columns of the input matrix.

The Transformer block $\text{TF}(\boldsymbol{X}) : \mathbb{R}^{d \times N} \to \mathbb{R}^{d \times N}$ is defined by:

$$\boldsymbol{X}' = \boldsymbol{X} + \text{Attn}(\boldsymbol{X}), \quad (3)$$

$$\text{TF}(\boldsymbol{X}) = \boldsymbol{X}' + \textbf{FF}(\boldsymbol{X}'). \quad (4)$$

where **FF** represent token-wise FF. In other words

$$\text{TF} = (\text{id} + \textbf{FF}) \circ (\text{id} + \text{Attn}), \quad (5)$$

where id denote the identity mapping. For simplicity, we omit the domain of definition.

For the analysis of expressive power in Section 3, we use the hardmax operator $\sigma_H$ instead of $\sigma_S$ and exclude Layer Normalizations as in previous studies (Yun et al., 2020; Kim et al., 2023).

### 2.2 LOOPED TRANSFORMER

Looped Transformers feed output back into input, defined as

$$\mathcal{L}_2 \circ \text{TF} \circ \cdots \circ \text{TF} \circ \mathcal{L}_1, \quad (6)$$

where $\mathcal{L}_2$ and $\mathcal{L}_1$ denote token-wise linear layers. Let $\text{TF}^{\circ r}$ denote the $r$-times composition of TF. We refer to $\mathcal{L}_2 \circ \text{TF}^{\circ r} \circ \mathcal{L}_1$ as a Looped Transformer with $r$-loops.

Looped Transformers have been studied in recent years, regarding their parameter efficiency (Lan et al., 2020; Takase & Kiyono, 2021; Bae et al., 2024) and generalization capabilities (Dehghani et al., 2019; Fan et al., 2024). Other recent works (Giannou et al., 2023; Gatmiry et al., 2024; Back De Luca & Fountoulakis, 2024; Gao et al., 2024; Giannou et al., 2024) have investigated their performance on iterative algorithms, including in-context learning and graph algorithm. In particular, Yang et al. (2024) empirically demonstrated that increasing the number of loop iterations enhances performance on complex tasks. However, to the best of our knowledge, there are no studies on the expressive power or approximation rate of Looped Transformers in function approximation.

### 2.3 THEORETICAL ANALYSIS ON FUNCTION APPROXIMATION

The universal approximation theorem for fully connected neural networks, as demonstrated by Cybenko (1989); Hornik et al. (1989), shows that networks of sufficient size can approximate certain classes of functions with arbitrarily low error. Transformers are universal approximators of sequence-to-sequence functions (Yun et al., 2020; Takakura & Suzuki, 2023; Jiang & Li, 2024; Kajitsuka & Sato, 2024), and their memorization capacity has also been studied (Kim et al., 2023). Recently, Zhang et al. (2023) revealed that even single fixed-size networks can be universal approximators. They explored the surprising potential of composition (loop) and derived the approximation rate in terms of the number of loop counts and modulus of continuity of the target function.

## 3 APPROXIMATION RATE OF LOOPED TRANSFORMERS

We establish the approximation rate of Looped Transformers by defining the modulus of continuity for continuous sequence-to-sequence functions. We begin with preliminaries of Transformers in Section 3.1. Then, we present and explain three types of continuity for sequence-to-sequence functions, which determine the approximation rate. In Section 3.3, we present our main results on approximation rate, along with some implications. In Section 3.4, we provide a proof sketch with a detailed explanation, outlining three steps for approximation and deriving the approximation rate.

### 3.1 PRELIMINARY

Transformers are *permutation-equivariant*, so we define the target function class as follows.

**Definition 3.1** (Yun et al. (2020); Kim et al. (2023)). A function $f : \mathbb{R}^{d \times N} \to \mathbb{R}^{d \times N}$ is said to be *permutation equivariant* if for any permutation matrix $\boldsymbol{P}$, we have $f(\boldsymbol{X}\boldsymbol{P}) = f(\boldsymbol{X})\boldsymbol{P}$. Let $\mathcal{F}_{\mathrm{PE}}([0,1]^{d \times N})$ denote the set of permutation equivariant and continuous functions.

To approximate sequence-to-sequence functions, networks need to map each token within the context of sequences, formulated as *contextual mapping*.

**Definition 3.2** (Yun et al. (2020); Kim et al. (2023)). Consider a finite set $\mathbb{L} \subset \mathbb{R}^{d \times N}$. A map $\mathrm{CM} : \mathbb{L} \to \mathbb{R}^{1 \times N}$ defines a *contextual mapping* if the map satisfies the following:

1. For any $\boldsymbol{L} \in \mathbb{L}$, the $N$ entries in $\mathrm{CM}(\boldsymbol{L})$ are all distinct.

2. For any $\boldsymbol{L}, \boldsymbol{L}' \in \mathbb{L}$, with $\boldsymbol{L} \neq \boldsymbol{L}'$, all entries of $\mathrm{CM}(\boldsymbol{L})$ and $\mathrm{CM}(\boldsymbol{L}')$ are distinct.

Let $\|\cdot\|_p$ denote the entry-wise $L^p$-norm for a vector for any $p \in [1, \infty)$.

**Definition 3.3** (Norm of function). We define the $L^p$-norm for a function $f$ on $[0,1]^{d \times N}$ by:

$$L^p([0,1]^{d \times N}) := \left( \int \|f(\boldsymbol{X})\|_p^p \, d\boldsymbol{X} \right)^{1/p}.$$

### 3.2 DEFINITION OF CONTINUITY FOR SEQUENCE-TO-SEQUENCE FUNCTIONS

The modulus of continuity of a continuous function $g : [0,1]^d \to \mathbb{R}$ can be defined as

$$\sup \left\{ |g(\boldsymbol{x}) - g(\boldsymbol{x}')| : \|\boldsymbol{x} - \boldsymbol{x}'\|_2 \leq t, \ \boldsymbol{x}, \boldsymbol{x}' \in [0,1]^d \right\}.$$

It can be extended for sequence-to-sequence functions $f : [0,1]^{d \times N} \to \mathbb{R}^{d \times N}$ as follows.

**Definition 3.4** (Modulus of Sentence Continuity). Given a sequence-to-sequence continuous function $f : [0,1]^{d \times N} \to \mathbb{R}^{d \times N}$, the modulus of *sentence continuity* is defined by:

$$\omega_f(t) := \sup \left\{ \|f(\boldsymbol{X}) - f(\boldsymbol{X}')\|_p : \|\boldsymbol{X} - \boldsymbol{X}'\|_2 \leq t, \ \boldsymbol{X}, \boldsymbol{X}' \in [0,1]^{d \times N} \right\}.$$

We illustrate what this continuity represents and why it is referred to as *sequence continuity*. If we consider the following two sentences: (1) I read books, and (2) He writes music, the sequence continuity measures how much the overall meaning of a sequence changes:

$$\text{'I read books'} \overset{\Delta}{\longleftrightarrow} \text{'He writes music'},$$

in proportion to the differences between the input sequences, measured by comparing each token:

$$\text{'I'} \overset{\Delta}{\longleftrightarrow} \text{'he'}, \qquad \text{'read'} \overset{\Delta}{\longleftrightarrow} \text{'write'}, \text{ and} \qquad \text{'book'} \overset{\Delta}{\longleftrightarrow} \text{'music'}.$$

We found that this concept is insufficient to derive the approximation rate of Looped Transformers, primarily because one key characteristic of the Transformer architecture is the sharing of parameters across all tokens: specifically, the feed-forward layers are applied token-wise. In other words, Transformers output token embeddings in the context of sequences for each token.

This observation leads us to define two additional forms of continuity: *contextual continuity* and *token continuity*, which we found to determine the approximation rate of Looped Transformers.

**Definition 3.5** (Modulus of Contextual Continuity). Given a sequence-to-sequence continuous function $f : [0,1]^{d \times N} \to \mathbb{R}^{d \times N}$, the modulus of *contextual continuity* is defined by:

$$\omega_f^{cont}(t) := \sup_{n, \boldsymbol{X}, \boldsymbol{X}'} \left\{ \|f(\boldsymbol{X})_{:,n} - f(\boldsymbol{X}')_{:,n}\|_p : \|\boldsymbol{X} - \boldsymbol{X}'\|_2 \le t, \ \boldsymbol{X}_{:,n} = \boldsymbol{X}'_{:,n}, \ \boldsymbol{X}, \boldsymbol{X}' \in [0,1]^{d \times N} \right\},$$

**Definition 3.6** (Modulus of Token Continuity). Given a sequence-to-sequence continuous function $f : [0,1]^{d \times N} \to \mathbb{R}^{d \times N}$, the modulus of *token continuity* is defined by:

$$\omega_f^{tok}(t) := \sup_{n, \boldsymbol{X}, \boldsymbol{X}'} \left\{ \|f(\boldsymbol{X})_{:,n} - f(\boldsymbol{X}')_{:,n}\|_p : \|\boldsymbol{X}_{:,n} - \boldsymbol{X}'_{:,n}\|_2 \le t, \right.$$

$$\left. \boldsymbol{X}_{:,m} = \boldsymbol{X}'_{:,m} \text{ for any } m \ne n, \ \boldsymbol{X}, \boldsymbol{X}' \in [0,1]^{d \times N} \right\},$$

The modulus of *contextual continuity* measures the variation in the output of token embeddings induced by a perturbation of context. For example, we consider the following three sentences:

(1) I write papers,      (2) You write books, and      (3) Mozart writes music.

The output embedding of the second token, 'write', should be similar in sentences (1) and (2) due to their similar context. In contrast, a larger variation in context, as seen in sentence (3), can induce a significant variation in the output of token embedding.

On the other hand, the modulus of *token continuity* measures the variation in the output embedding caused by perturbations to the token itself within the same context. For instance, we consider the sentences:

(1) I write papers, and      (2) I draft papers.

In this example, both sentences have the same context, but the verb ('write' vs. 'draft') variation reflects a perturbation in the token itself. The modulus of *token continuity* quantifies how this change influences the output embeddings. A small modulus of *token continuity* means that the output embeddings of 'write' and 'draft' are expected to be similar.

## 3.3 MAIN RESULT

The following main theorem demonstrates the approximation rate of Looped Transformers in terms of the modulus of continuity and the number of loops.

**Theorem 3.7.** *Given a function $f \in \mathcal{F}_{PE}([0,1]^{d \times N})$, for any $r \in \mathbb{N}$, there exists a Looped Transformer* $\mathrm{TF} : \mathbb{R}^{(24d+1) \times N} \to \mathbb{R}^{(24d+1) \times N}$ *of single head, head size $s = 1$, and width size $q = 99d + 8$, and two affine linear maps $\mathcal{L}_1 : \mathbb{R}^d \to \mathbb{R}^{24d+1}$ and $\mathcal{L}_2 : \mathbb{R}^{24d+1} \to \mathbb{R}^d$ such that*

$$\left\| \boldsymbol{\mathcal{L}}_2 \circ \mathrm{TF}^{\circ r} \circ \boldsymbol{\mathcal{L}}_1 - f \right\|_{L^p([0,1]^{d \times N})} \le \omega_f^{tok}(\delta\sqrt{d}) + \omega_f^{cont}(\delta\sqrt{Nd}) + \omega_f(\delta\sqrt{Nd}) + \mathcal{O}(\delta^d),$$

*for $\delta = \big((r - N)/2\big)^{-1/((N+1)d+1)}$, where $\boldsymbol{\mathcal{L}}_1$ and $\boldsymbol{\mathcal{L}}_2$ represent the token-wise applications of $\mathcal{L}_1$ and $\mathcal{L}_2$, respectively.*

Thus, Looped Transformers are universal approximators.

**Corollary 3.8.** *The hypothesis space of Looped Transformers $\mathcal{H}$, defined by*

$$\mathcal{H} := \left\{ \boldsymbol{\mathcal{L}_2} \circ \mathrm{TF}^{\circ r} \circ \boldsymbol{\mathcal{L}_1} : r \in \mathbb{N}, \ \boldsymbol{\mathcal{L}_2} \text{ and } \boldsymbol{\mathcal{L}_1} \text{are token-wise affine linear maps} \right\},$$

*are dense in $\mathcal{F}_{PE}([0,1]^{d \times N})$ in terms of the $L^p([0,1]^{d \times N})$ norm.*

These results provide us with some insights:

- A function with a small modulus of continuities, *e.g.*, robust to contextual perturbations, is suited for approximation by Looped Transformers.
- The total parameter count is $\mathcal{O}(d)$, independent of both $\delta$ and $N$, highlighting the parameter efficiency of Looped Transformers.
- The optimal approximation rate of ReLU networks of size $n$ is $\mathcal{O}(\omega_f(\mathcal{O}(n^{-2/d})))$ for continuous functions on $[0,1]^d$ (Yarotsky, 2018); the exponential rate is unavoidable.

## 3.4 PROOF SKETCH

We provide a proof sketch highlighting differences from prior studies and associated difficulties. The formal proof is provided in Appendix A.

**Approximation with Piecewise Constant Function.** We approximate $f \in \mathcal{F}_{\mathrm{PE}}$ with piece-wise constant function $\bar{f} : [0,1]^{d \times N} \to \mathbb{R}^{d \times N}$. Specifically, for $\delta^{-1} \in \mathbb{N}$, we divide the input space $[0,1]^{d \times N}$ into $\delta$-discretized cubes, denoted by $\{Q_{\boldsymbol{\mathcal{B}}}\}_{\boldsymbol{\mathcal{B}} \in \{0,1,\dots,\delta^{-1}-1\}^{d \times N}}$. Each cube is associated with a representative $\hat{\boldsymbol{X}}_{\boldsymbol{\mathcal{B}}} \in Q_{\boldsymbol{\mathcal{B}}}$. Define a piecewise constant function $\bar{f}$ for $\boldsymbol{X} \in [0,1]^{d \times N}$ as

$$\bar{f}(\boldsymbol{X}) = f(\hat{\boldsymbol{X}}_{\boldsymbol{\mathcal{B}}}) \quad \text{where } \boldsymbol{\mathcal{B}} \text{ satisfies } \boldsymbol{X} \in Q_{\boldsymbol{\mathcal{B}}}.$$

We can bound the approximation error within each cube as $\|\bar{f}(\boldsymbol{X}) - f(\boldsymbol{X})\|_p \leq \omega_f(\sqrt{\delta^2 + \dots + \delta^2}) \leq \omega_f(\delta \cdot \sqrt{Nd})$ for any $\boldsymbol{X} \in [0,1]^{d \times N}$. We involve three steps to construct $\bar{f}$. The first and second steps map the input $\boldsymbol{X}$ to the coordinates of the discretized input space, involving $\boldsymbol{\mathcal{B}}$. The third step approximately maps these coordinates to the target embeddings.

**Step 1. Token-wise Quantization.** The network, with $\delta^{-1} - 1$ loops, token-wise maps the input space into indices (the proof is provided in Appendix A.4). Then it maps them to an integer, referred to as a *token ID*:

$$\boldsymbol{X}_{:,n} \in [0,1]^d \to \boldsymbol{\beta} \in \{0,1,\dots,\delta^{-1}-1\}^d \to z \in \{0,1,\dots,\delta^{-d}-1\}. \tag{7}$$

The key idea behind our proof follows Zhang et al. (2023); however, we cannot directly apply it here due to the need to account for skip connections. Additionally, it is necessary to consider a bijective mapping of $\boldsymbol{\beta}$ to a token ID in the $\delta^{-1}$-base system for the next step.

**Step 2. Contextual Mapping.** The network performs contextual mapping, which maps $N$ token IDs to a *sequence ID* in the set of $\{0,1,\dots,\delta^{-Nd}-1\}$. Previous studies (Yun et al., 2020; Kim et al., 2023) use multiple layers for constructions; however, these results do not extend to Looped Transformers for two reasons.

(1) Yun et al. (2020) used both sparse and uniform attention, whereas Looped Transformers are limited to a single fixed attention layer.

(2) Kim et al. (2023) used $N$-layers to store $N$ parameters, whereas fixed-size $\mathcal{O}(d)$ Looped Transformers cannot store $N$ weight components.

While these considerations indicate some limitations of Looped Transformers, we found that Looped Transformers with $N$-loops can perform contextual mapping. The proof strategy follows Kim et al. (2023); however, it is necessary to update for a single Transformer block. Let $\boldsymbol{z} \in \{0,1,\dots,\delta^{-d}-1\}^N$ represent a sequence of $N$ ordered and distinct token IDs, where $\boldsymbol{z}_1 > \boldsymbol{z}_2 > \dots > \boldsymbol{z}_N$. The networks map the set of token IDs into sequence ID via inner product with $\boldsymbol{u} := (\delta^{-d(N-1)},\dots,\delta^{-d},1) \in \mathbb{R}^N$ *i.e.*

$$\mathrm{CM}(\boldsymbol{z}) := \boldsymbol{u}^\top \boldsymbol{z},$$

which satisfy

$$\left\| \boldsymbol{u}^\top \boldsymbol{z} - \boldsymbol{u}^\top \boldsymbol{z}' \right\| > 1, \quad \text{if } \boldsymbol{z} \neq \boldsymbol{z}'.$$

Thus CM is a contextual mapping. The key point is that the network only needs to store $\delta$ to represent $\boldsymbol{u}$, allowing it to be implemented with Looped Transformers. Details are provided in Appendix A.5

**Step 3. Token-wise Mapping.** The network token-wise maps the coordinates of discretized regions approximately to the target token embedding. From Steps 1 and 2, each token in the input sequence is assigned a token ID with a sequence ID, where the sequence ID is consistent across all tokens. The combination of the token ID and sequence ID determines the coordinates, referred to as *contextual token ID*.

Notably, we found that the design of the *contextual token ID* plays a crucial role in Looped Transformers. This comes from the constraint of looped architecture. Let $\mathcal{K}$ denote the set of contextual

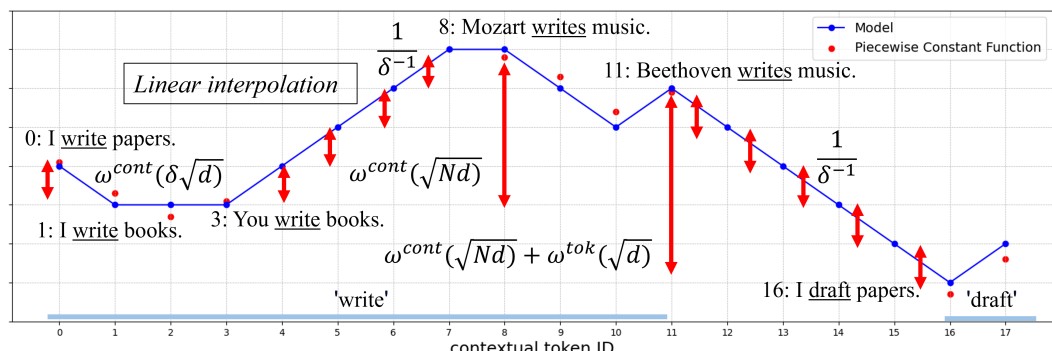

Figure 1: Approximation error and modulus of continuity. The linear interpolation technique reduces the error by a factor of $1/\delta^{-1}$.

token IDs, where each index is associated with a distinct cube $\mathcal{B} \in \{0, 1, \ldots, \delta^{-1} - 1\}^{d \times N}$ in the discretized space. For each $k \in \mathcal{K}$, let $\boldsymbol{X}_k$ denote the cube associated with $\mathcal{B}$, *i.e.*, $\boldsymbol{X}_{\mathcal{B}}$. Looped Transformer $\tilde{f} : \mathbb{R}^{d \times N} \to \mathbb{R}^{d \times N}$ can token-wise approximate piecewise constant function $\bar{f}$ with the error of

$$\|\tilde{f}(\boldsymbol{X}_k)_{:,n} - \bar{f}(\boldsymbol{X}_k)_{:,n}\| \leq \max_{n, k' \in \mathcal{K}} \|\bar{f}(\boldsymbol{X}_{k'})_{:,n} - \bar{f}(\boldsymbol{X}_{k'-1})_{:,n}\|_p \quad \text{for any } k \in \mathcal{K}, \tag{8}$$

for $n = 0, 1, \ldots, N$ (details are provided in Lemma 4.1). This requires us to design $\mathcal{K}$ so that $\|\bar{f}(\boldsymbol{X}_{k'})_{:,n} - \bar{f}(\boldsymbol{X}_{k'-1})_{:,n}\|_p$ is small, *i.e.*, the outputs of neighboring contextual token IDs are similar. The core idea of this design is explained with illustrations, comparing the output embeddings of the underlined tokens.

(1) I write papers. ; I write papers.   (different token ID with same sequence ID)

(2) I write papers. ; You write books.   (same token ID with different sequence ID)

The contextual continuity $\omega_f^{cont}$ of Definition 3.2 ensures that the outputs embeddings of 'write' in (2) are similar. However, none of the continuity properties provide guarantees that the output embeddings of 'write' and 'papers' in (1) are similar. Thus, we design the contextual token ID so that the same token with different sequence IDs comes next to each other except for unavailable corner cases (details in Appendix A.4).

**Consolidation into Single Looped Transformer.**   At the end of the construction, we demonstrate that the composition of the three sub-networks from Steps 1, 2, and 3 can be realized with a single Transformer block. The proof strategy follows Zhang et al. (2023); however, it cannot be directly applied because their approach requires an additional layer. In contrast, we found that a single Transformer block is sufficient (details are provided in Appendix A).

**Deriving Approximation Rate.**   Lastly, we estimate the approximation error of our construction and establish the approximation rate. In Step 2, we consider only the case where all $N$ input tokens are distinct, disregarding other cases. These cases can be treated as negligible when $\delta$ is small. The number of subsets where one of the $N$ tokens is duplicated is

$$(\delta^{-d})^N - \delta^{-d} \cdot (\delta^{-d} - 1) \cdots (\delta^{-d} - N - 1) < C\delta^{-(N-1)d},$$

where $C$ is a constant. The volume of these subsets is $C\delta^{-(N-1)d}/\delta^{-Nd} = C\delta^d$, thus the error with respect to the $L^p$ norm is $\mathcal{O}(\delta^d)$.

In Step 3, we can bound the approximation error as follows:

$$\|\tilde{f}(\boldsymbol{X}_k) - \bar{f}(\boldsymbol{X}_k)\|_p \leq \max_n \|\tilde{f}(\boldsymbol{X}_k)_{:,n} - \bar{f}(\boldsymbol{X}_k)_{:,n}\|_p \leq \max_{n, k' \in \mathcal{K}} \|\bar{f}(\boldsymbol{X}_{k'})_{:,n} - \bar{f}(\boldsymbol{X}_{k'-1})_{:,n}\|_p.$$

There are two types of error for the right-hand side term: the variation induced by contextual perturbation and the variation induced by token perturbation. We illustrate this with examples of each pattern, as shown in Fig. 1. Specifically, we consider the following three cases:

(1) I write papers. ; I write books. (*small* perturbation of context)

(2) You write books. ; Mozart write music. (*large* perturbation of context)

(3) Beethoven writes music. ; I draft papers. (perturbation of both token and context)

The error in each case can be bounded with the corresponding modulus of continuity:

1. $\omega_f^{cont}(\delta\sqrt{d})$

2. $\omega_f^{cont}(\sqrt{Nd}) \to \delta\omega_f^{cont}(\sqrt{Nd})$

3. $\omega_f^{tok}(\sqrt{d}) + \omega_f^{cont}(\sqrt{Nd}) \to \delta\left(\omega_f^{tok}(\sqrt{d})\right) + \omega_f^{cont}(\sqrt{Nd})\right)$

where $\to$ represents the use of *linear interpolation techniques* to reduce the error with extra $\delta$-loops (explained in Appendix A). Since that $\omega_f^{cont,tok}(n\cdot t) \le n\cdot\omega_f^{cont,tok}(t)$ for any $n \in \mathbb{N}$ and $t \in [0,\infty)$ with $\delta < 1$, we can then derive the upper bound for the three terms:

$$
\begin{aligned}
\max_{n,k'\in\mathcal{K}} &\|\bar{f}(\boldsymbol{X}_{k'})_{:,n} - \bar{f}(\boldsymbol{X}_{k'-1})_{:,n}\|_p \\
&\le \max\left\{\omega_f^{cont}(\delta\sqrt{d}),\ \delta\omega_f^{cont}(\sqrt{Nd}),\ \delta\left(\omega_f^{tok}(\sqrt{d}) + \omega_f^{cont}(\sqrt{Nd})\right)\right\} \\
&\le \max\left\{\omega_f^{cont}(\delta\sqrt{d}),\ \omega_f^{cont}(\delta\sqrt{Nd}),\ \omega_f^{tok}(\delta\sqrt{d}) + \omega_f^{cont}(\delta\sqrt{Nd})\right\} \\
&\le \omega_f^{tok}(\delta\sqrt{d}) + \omega_f^{cont}(\delta\sqrt{Nd}),
\end{aligned}
$$

With the triangle inequality, we have an approximation error as

$$
\begin{aligned}
\|\tilde{f} - f\|_{L^p([0,1]^{d\times N})} &\le \|\tilde{f}(\boldsymbol{X}) - f(\boldsymbol{X})\|_p \cdot 1 \\
&\le \|\tilde{f}(\boldsymbol{X}) - \bar{f}(\boldsymbol{X})\|_p + \|\bar{f}(\boldsymbol{X}) - f(\boldsymbol{X})\|_p \\
&\le \max_{n,k'\in\mathcal{K}}\|\bar{f}(\boldsymbol{X}_{k'})_{:,n} - \bar{f}(\boldsymbol{X}_{k'-1})_{:,n}\|_p + \|\bar{f}(\boldsymbol{X}) - f(\boldsymbol{X})\|_p + \mathcal{O}(\delta^d) \\
&\le \omega_f^{tok}(\delta\sqrt{d}) + \omega_f^{cont}(\delta\sqrt{Nd}) + \omega_f(\delta\sqrt{Nd}) + \mathcal{O}(\delta^d). \quad (9)
\end{aligned}
$$

Then, $\delta$ is expressed in terms of the number of loops $r$ to determine the approximation rate. We use $\delta^{-1} - 1$ loops for Step 1, $N$ loops for Step 2, and $2\delta^{-(N+1)d} - 1$ loops for Step 3, with 1 loop required to connect each step. Thus we have

$$
\begin{aligned}
r = (\delta^{-1} - 1) + 1 + (N) + 1 + \left(2\delta^{-(N+1)d} - 1\right) &\Leftrightarrow \delta^{-1} + 2\delta^{-(N+1)d} = r - N \\
&\Leftrightarrow \delta^{-1} \cdot 2\delta^{-(N+1)d} \ge r - N \\
&\Leftrightarrow 2\delta^{-(N+1)d-1} \ge r - N \\
&\Leftrightarrow \delta \le \left(\frac{r-N}{2}\right)^{-1/((N+1)d+1)}. \quad (10)
\end{aligned}
$$

From Eq. 9 and Eq. 10, we can derive Theorem 3.7.

**Summary.** Our contribution is to establish an approximation rate for Looped Transformers by identifying the continuity of sequence-to-sequence functions. Additionally, as a technical contribution, we demonstrate that a single Looped Transformer block is sufficient for contextual mapping. While Zhang et al. (2023) requires three feed-forward layers of looped ReLU networks for universal approximation, we achieve this with just one layer.

## 4 FROM THEORY TO PRACTICE: INTRODUCING TIMESTEP ENCODING

The theoretical result in Section 3 identifies a limitation of the looped architecture in its reliance on contextual and token continuity. This analysis suggests incorporating *time-dependent* scaling parameters for each loop, which we implemented as a function of the timestep encoding.

## 4.1 MOTIVATION

**Limitation Specific to Looped Architecture.** Theorem 3.7 shows that the approximation rate of Looped Transformers depends on the modulus of three types of continuity. Sequence continuity relates to approximating continuous functions with piecewise constants, while contextual and token continuity dependencies are unique to the looped architecture. Previous studies (Yun et al., 2020; Kim et al., 2023) show that standard Transformers lack these dependencies. This additional dependency increases approximation errors, limiting the approximation power of Looped Transformers.

We identify the cause of dependency in **Step 3: Token-wise Mapping** of the following Lemma.

**Lemma 4.1.** *Given* $\boldsymbol{y}_k \in \mathbb{R}^d$ *for* $k = 0, 1, \dots, m - 1$ *with*

$$|(\boldsymbol{y}_k - \boldsymbol{y}_{k-1})_i| \le \varepsilon_i \quad \text{for } k = 1, 2, \dots, m - 1,$$

*there exist feed-forward layer* $\mathrm{FF} : \mathbb{R}^{14d} \to \mathbb{R}^{14d}$ *of width size* $20d$ *and two affine linear maps* $\mathcal{L}_1 : \mathbb{R}^d \to \mathbb{R}^{14d}$ *and* $\mathcal{L}_2 : \mathbb{R}^{14d} \to \mathbb{R}^d$ *such that*

$$\left|\left(\mathcal{L}_2 \circ (\mathrm{id} + \mathrm{FF})^{(m-1)} \circ \mathcal{L}_1(k) - \boldsymbol{y}_k\right)_i\right| \le \varepsilon_i \quad \text{for } k = 0, 1, \dots, m - 1,$$

*for any* $i = 1, 2, \dots, d$.

Lemma 4.1 implies that large variations in the target function, represented by discretized points $\boldsymbol{y}_k$, lead to increased approximation error in Looped Transformers. Specifically, when outputs at nearby points vary greatly, a small approximation error cannot be guaranteed.

**How Can We Improve the Approximation Rate of Looped Transformers?** To address the dependency on contextual and token continuity, we introduce *time-dependent* parameters for each loop. Specifically, we modify the feed-forward layers by adding a scaling vector parameter that varies with the loop index (timestep), defined as follows:

$$\mathrm{FF}(\boldsymbol{X}) \to \boldsymbol{\eta}(t) \odot \mathrm{FF}(\boldsymbol{X}) \quad \text{for the } t\text{-th loops},$$

where $\odot$ is an element-wise product, $t \in \mathbb{N}$ denotes the loop index (timestep), and $\boldsymbol{\eta}(t) \in \mathbb{R}^d$ represents a *time-dependent* scaling parameter. This kind of *dynamic* scaling vector parameters is also used by HyperNetworks (Ha et al., 2016) for RNN to enhance expressive power.

We show that the time-dependent Looped Transformer overcomes approximation errors from contextual and token continuity. Specifically, we can replace Lemma 4.1 with the following Theorem 4.2, which demonstrates that time-dependent models can precisely approximate any target function. The proof is provided in Appendix B.

**Theorem 4.2.** *Given* $\boldsymbol{y}_k \in \mathbb{R}^d$ *for* $k = 0, \dots, m - 1$, *there exist feed-forward layer* $\mathrm{FF} : \mathbb{R}^{4d} \to \mathbb{R}^{4d}$ *of width size* $6d$ *and* $\boldsymbol{\eta}(t) \in \mathbb{R}^{4d}$ *and two affine linear maps* $\mathcal{L}_1 : \mathbb{R}^d \to \mathbb{R}^{4d}$ *and* $\mathcal{L}_2 : \mathbb{R}^{4d} \to \mathbb{R}^d$ *s.t.*

$$\mathcal{L}_2 \circ (\mathrm{id} + \boldsymbol{\eta}(m-1) \odot \mathrm{FF}) \circ \cdots \circ (\mathrm{id} + \boldsymbol{\eta}(1) \odot \mathrm{FF}) \circ \mathcal{L}_1(k) = \boldsymbol{y}_k.$$

For implementation, while adding parameters for each loop is effective, the number of parameters increases with the number of loops. Therefore, we introduce timestep encoding to address this issue.

## 4.2 TIMESTEP ENCODING

We use timestep encodings to represent loop counts and to condition scaling parameters, following Peebles & Xie (2023), where *time-dependent* Transformers are applied in diffusion models by regressing layer normalization parameters from timestep encodings.

To condition on timesteps, frequency embeddings are processed through a two-layer MLP with hidden size matching the Transformer block and SiLU activation, as shown in Fig. 2. Let $\mathrm{TE}(t) \in \mathbb{R}^d$ denote timestep embeddings, defined as:

$$\mathrm{TE}(t) = \boldsymbol{W}_1^{(TE)} \cdot \mathrm{SiLU}(\boldsymbol{W}_2^{(TE)} \cdot \mathrm{PE}(t) + \boldsymbol{b}_1^{(TE)}) + \boldsymbol{b}_2^{(TE)},$$

where $\boldsymbol{W}_1^{(TE)}, \boldsymbol{W}_2^{(TE)} \in \mathbb{R}^{d \times d}$ and $\boldsymbol{b}_1^{(TE)}, \boldsymbol{b}_2^{(TE)} \in \mathbb{R}^d$ are parameters, and $\mathrm{PE}(t) \in \mathbb{R}^d$ is the timestep encoding function that maps the timestep (loop index) into a $d$-dimensional embedding, *s.t.*

$$\mathrm{PE}(t)_{2i} = \sin(t/10000^{2i/d}), \quad \mathrm{PE}(t)_{2i+1} = \cos(t/10000^{2i/d}).$$

The root mean square layer normalization (RMSNorm) (Zhang & Sennrich, 2019) has been employed in several recent large language models (LLMs), such as LLaMA (et al., 2023) and Gemma (Team, 2024). RMSNorm, denoted by $\mathrm{RMSLN}$, is defined as

$$\mathrm{RMSLN}(\boldsymbol{x}) = \boldsymbol{\alpha} \odot \frac{\boldsymbol{x}}{\mathrm{RMS}(\mathbf{x})}, \quad \text{where} \ \ \mathrm{RMS}(\mathbf{x}) = \sqrt{\frac{1}{d}\sum_{i=1}^{d} \boldsymbol{x}_i^2}.$$

where $\boldsymbol{\alpha} \in \mathbb{R}^d$ is a gain parameter for rescaling.

Extending standard RMSNorm, we define time-dependent RMSNorm as:

$$\mathrm{RMSLN}(\boldsymbol{x}, t) = \boldsymbol{\alpha}(t) \odot \frac{\boldsymbol{x}}{\mathrm{RMS}(\mathbf{x})}$$

Figure 2: Timestep encoding architecture.

where $\boldsymbol{\alpha}(t) \in \mathbb{R}^d$ is a time-dependent parameter generated by a *hypernetowrk*.Additionally, we incorporate parameters for output scaling, defining the time-dependent Transformer block as follows:

$$\boldsymbol{X}' = \boldsymbol{X} + \boldsymbol{\gamma}_1(t) \odot \mathrm{Attn}(\mathbf{RMSLN_1}(\boldsymbol{X}, t)),$$

$$\mathrm{TF}(\boldsymbol{X}, t) = \boldsymbol{X}' + \boldsymbol{\gamma}_2(t) \odot \mathbf{FF}(\mathbf{RMSLN_2}(\boldsymbol{X}', t)),$$

where $\boldsymbol{\gamma}_1(t), \boldsymbol{\gamma}_2(t) \in \mathbb{R}^d$ are time-dependent parameters applied token-wise, and $\mathbf{RMSLN_1}$ and $\mathbf{RMSLN_2}$ represent token-wise applications of $\mathrm{RMSLN_1}$ and $\mathrm{RMSLN_2}$, respectively.

To generate time-dependent vector parameters, we use the SiLU function and weight parameters:

$$\boldsymbol{\alpha}_1(t), \boldsymbol{\alpha}_2(t), \boldsymbol{\gamma}_1(t), \boldsymbol{\gamma}_2(t) = \boldsymbol{W}^{(H)} \cdot \mathrm{SiLU}(\mathrm{TE}(t)) + \boldsymbol{b}^{(H)},$$

where $\boldsymbol{W}^{(H)} \in \mathbb{R}^{4d \times d}, \boldsymbol{b}^{(H)} \in \mathbb{R}^d$ are parameters.

## 5 EXPERIMENTS

Our experimental results support our theoretical findings. First, we show that increasing loop counts enhances the expressive power of Looped Transformers by evaluating dynamic programming (DP) tasks in Section 5.1. Second, we observe performance gains from timestep encodings in certain DP, in-context learning, and language modeling tasks Section 5.2.

### 5.1 VARYING LOOPS WITH DYNAMIC PROGRAMMING

DP problems were chosen for their recursive structure and their difficulty for standard Transformers without chain-of-thought (Feng et al., 2023). We categorize specific types of DP problems and select representative tasks from each category (details are provided in Appendix C.1).

**Experimental setups.** We generate $10^6$ samples for training and $10^3$ samples for testing. All tasks are trained as classification tasks using cross-entropy loss and are evaluated by best accuracy on the test sets. While our theoretical results focus on approximation power (fitting to the training set), we observe a strong correlation between training and test accuracy, suggesting that lower test accuracy reflects the approximation capacity rather than a generalization issue (see the results of ED in Appendix C.2).

We trained Looped Transformers with 5, 10, 50, and 100 loops, both with and without time dependency (the configuration is provided in Appendix C.2. We incrementally increased the number of loops, stopping when performance exceeded 90% or saturated due to limited computational resources. In addition, to validate the research significance of Looped Transformers, we compared them to a 12-layer Transformer. The Transformer's limitations stem not from approximation but from generalization capabilities, which fall outside the scope of our theoretical results.

**Results.** Certain tasks require a large number of loops, while others benefit from fewer iterations (see Table. 1). We observe accuracy improvements for both LCS and ED tasks by increasing the number of loops. In particular, for ED tasks, models with timestep encoding exhibit consistent accuracy gains without saturation as the loop count grows. Moreover, Looped Transformers solve tasks that standard Transformers cannot.

Table 1: Test accuracy for dynamic programming (DP) tasks, with parameters (e.g., sequence length) indicated in parentheses. We limit training and evaluation to lower loop counts if tasks are effectively solved with fewer loops. Some tasks require high loop counts, while others are solvable with few.

| Task | TF | Looped TF | | | | w/ Timestep Encoding | | | |
|------|------|------|------|------|------|------|------|------|------|
| | d=12 | r=5 | r=10 | r=50 | r=100 | r=5 | r=10 | r=50 | r=100 |
| Subset Sum (10) | 83.4 | **84.1** | 83.0 | - | - | 83.8 | 83.9 | - | - |
| Knapsack (20) | 92.8 | 92.2 | **94.0** | - | - | 88.7 | 90.9 | - | - |
| LCS (60) | 70.0 | 66.0 | 81.8 | 98.6 | - | 68.5 | 80.5 | **99.3** | - |
| LCS (100) | 39.8 | 39.6 | 45.1 | 93.5 | - | 36.7 | 45.6 | **98.1** | - |
| ED (60) | 41.4 | 23.8 | 32.6 | 47.3 | 47.7 | 26.6 | 38.9 | 57.3 | **88.3** |

## 5.2 ENHANCEMENT VIA TIMESTEP ENCODING

**In-Context Learning.** Transformers can learn in-context (Brown, 2020), with recent studies examining their ability to learn function classes (Garg et al., 2022; Akyürek et al., 2023; Von Oswald et al., 2023). Yang et al. (2024) investigated with Looped Transformers. We evaluate timestep encodings with 12-loops on decision tree functions, as described in Appendix C.3. We observe enhancement via timestep encodings (Table 2), surpassing even standard Transformers with 12 layers.

Table 2: MSE ($\downarrow$) on in-context learning results with enhancement via timestep encoding.

| TF d=12 | Looped r=12 | Timestep r=12 |
|---------|-------------|---------------|
| 8.64e-03 | 1.43e-02 | **1.70e-03** |

Table 3: Enhancement via timestep encoding on perplexity performance for WikiText-103.

| Metric | TF | Looped TF | | | w/ Timestep Encoding | | |
|--------|------|------|------|------|------|------|------|
| | d=12 | r=1 | r=3 | r=6 | r=1 | r=3 | r=6 |
| Train Perplexity ($\downarrow$) | 5.11 | 6.65 | 5.64 | 5.61 | 6.29 | 5.31 | **5.05** |
| Test Perplexity ($\downarrow$) | **19.6** | 33.11 | 27.93 | 28.16 | 31.18 | 23.45 | 22.42 |

**Language Modeling.** We use the WikiText-103 (Merity et al., 2017) dataset, containing over 100 million tokens from Wikipedia articles, to compare wide-block Looped Transformers, approximately matched in parameters to 12-layer standard Transformers, with and without timestep encodings across 1, 3, and 6 loops, evaluated by perplexity. Details are in Appendix C.4. We observe that timestep encoding enhances training approximation and improves test perplexity (Table 3). We also found that test perplexity falls short compared to standard Transformer.

## 6 CONCLUSION

We have established approximation rates for Looped Transformers by introducing the modulus of continuity for sequence-to-sequence functions. Our analysis reveals a limitation inherent to the looped architecture, prompting the incorporation of a time-dependent scaling parameter. This research is the first to investigate the function approximation capabilities of Looped Transformers. Additionally, our experiments demonstrate that Looped Transformers can solve certain dynamic programming (DP) tasks that traditional Transformers struggle with, though the underlying mechanism remains unclear. While we have derived upper bounds, the tightness of these approximation rates is still undetermined. Future work will explore more complex and practical tasks, such as mathematical problem-solving.

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

## A PROOFS OF FUNCTION APPROXIMATION

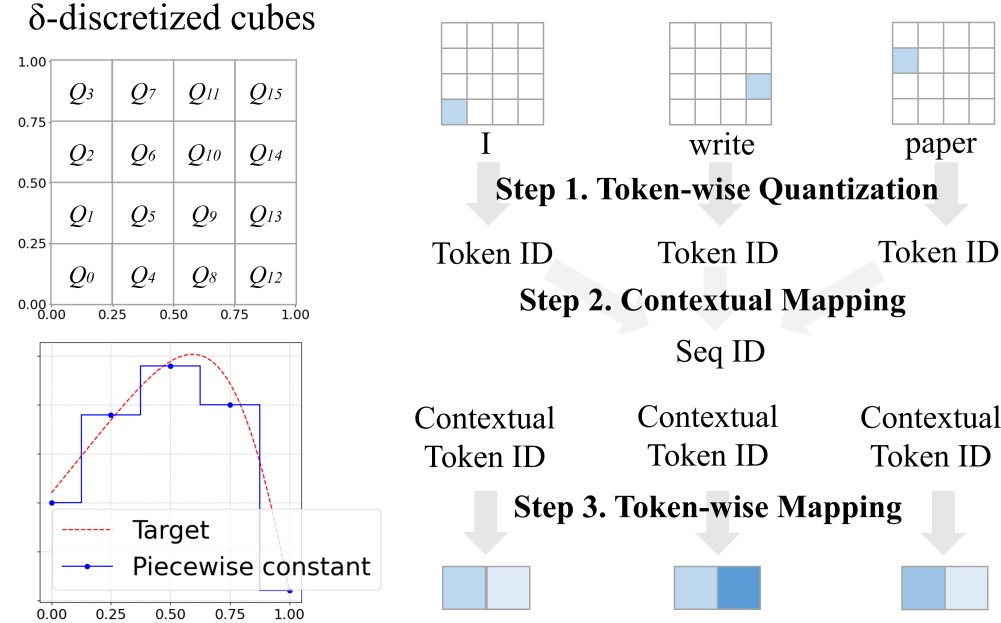

Figure 3: Overview of proof sketch.

We present the formal proof of Theorem 3.7. In Appendix A.2, we provide the whole proof relying on lemmas, introduced in the following subsections. In Appendix A.4, Appendix A.5, and Appendix A.6, we provide the key lemmas. The overview of proof outline is shown in Fig 3.

### A.1 NOTATIONS

- The bold notation for functions indicates that they are applied in a token-wise manner.
- We define the nearest functions as follows:

$$\text{near}^+(a, \mathcal{S}) \coloneqq \underset{b \in \mathcal{S}, b > a}{\arg\min} |a - b|$$

$$\text{near}^-(a, \mathcal{S}) \coloneqq \underset{b \in \mathcal{S}, b < a}{\arg\min} |a - b|$$

The function $\text{near}(a, \mathcal{S})$ identifies the element in the set $\mathcal{S}$ that is closest to $a$. The function $\text{near}^+(a, \mathcal{S})$ finds the closest element greater than $a$, while $\text{near}^-(a, \mathcal{S})$ identifies the closest element less than $a$.

- Given two values $a$ and $b$, and the number of divisions $n$, we can define the linear interpolation output at an index $t$ (where $t$ ranges from $0$ to $n$) as follows:

$$\text{lin\_interp}(a, b, t, n) := a + \frac{t}{n}(b - a)$$

## A.2 PROOF OF THEOREM 3.7

**Theorem 3.7.** *Given a function $f \in \mathcal{F}_{\text{PE}}([0,1]^{d \times N})$, for any $r \in \mathbb{N}$, there exists a Looped Transformer* $\text{TF} : \mathbb{R}^{(24d+1) \times N} \to \mathbb{R}^{(24d+1) \times N}$ *of single head, head size $s = 1$, and width size $q = 99d + 8$, and two affine linear maps $\mathcal{L}_1 : \mathbb{R}^d \to \mathbb{R}^{24d+1}$ and $\mathcal{L}_2 : \mathbb{R}^{24d+1} \to \mathbb{R}^d$ such that*

$$\left\| \boldsymbol{\mathcal{L}}_2 \circ \text{TF}^{\circ r} \circ \boldsymbol{\mathcal{L}}_1 - f \right\|_{L^p([0,1]^{d \times N})} \leq \omega_f^{tok}(\delta\sqrt{d}) + \omega_f^{cont}(\delta\sqrt{Nd}) + \omega_f(\delta\sqrt{Nd}) + \mathcal{O}(\delta^d),$$

*for $\delta = \big((r - N)/2\big)^{-1/((N+1)d+1)}$, where $\boldsymbol{\mathcal{L}}_1$ and $\boldsymbol{\mathcal{L}}_2$ represent the token-wise applications of $\mathcal{L}_1$ and $\mathcal{L}_2$, respectively.*

*Proof.* Since any continuous function can be approximated by a piecewise constant function with arbitrarily small errors, we approximate $f \in \mathcal{F}_{\text{PE}}$ with piece-wise constant function $\bar{f} : [0,1]^{d \times N} \to \mathbb{R}^{d \times N}$. We choose $\delta^{-1} \in \mathbb{N}$, which determines how finely the input is divided; then, we divide the input space $[0,1]^{d \times N}$ into $\delta$-discretized cubes, denoted by $\{Q_{\boldsymbol{\mathcal{B}}}\}$ for $\boldsymbol{\mathcal{B}} \in \{0, 1, \ldots, \delta^{-1} - 1\}^{d \times N}$ defined by

$$Q_{\boldsymbol{\mathcal{B}}} := \Big\{ \mathbf{X} \in [0,1]^{d \times N} : \mathbf{X}_{i,n} \in \big[\boldsymbol{\mathcal{B}}_{i,n}\delta, \boldsymbol{\mathcal{B}}_{i,n}\delta + 1\big), \quad i = 1, 2, \ldots, Nd \Big\}.$$

Note that we do not consider the *trifling regions* in Zhang et al. (2023) as we use the $L_p$ norm rather than the uniform norm.

Each cube is associated with representative $\hat{\mathbf{X}}_{\boldsymbol{\mathcal{B}}} \in Q_{\boldsymbol{\mathcal{B}}}$. We can define piecewise constant function $\bar{f}$ for $\mathbf{X} \in [0,1]^{d \times N}$ as

$$\bar{f}(\mathbf{X}) := f(\hat{\mathbf{X}}_{\boldsymbol{\mathcal{B}}}) \quad \text{where } \boldsymbol{\mathcal{B}} \text{ satisfies } \mathbf{X} \in Q_{\boldsymbol{\mathcal{B}}}.$$

We can bound the error within each cube as: $\|\bar{f}(\mathbf{X}) - f(\mathbf{X})\|_p \leq \omega_f(\delta\sqrt{Nd})$ for any $\mathbf{X} \in [0,1]^{d \times N}$.

Our construction consists of three steps to approximate $\bar{f}$. The first and second steps map the input $\mathbf{X}$ to the coordinates of the discretized input space. The third step approximately maps these coordinates to the target embeddings. The outline of three step is:

1. The network, with $(\delta^{-1} - 1)$-loops, maps the input space $[0,1]^d$ token-wise to the coordinates $\boldsymbol{\beta} \in \{0, 1, \ldots, \delta^{-1} - 1\}^d$ of divided cubes, and then bijectively maps these coordinates to an integer, representing token IDs in the set $\{0, 1, \ldots, \delta^{-d}\}$, using a $\delta^{-1}$-base system.

2. The network, with $N$ loops, performs a contextual mapping from the set of distinct $N$ token IDs into the set of *extended sequence ID*. Specifically, the network using Step1 and Step2 maps the discretized coordinates $\boldsymbol{\mathcal{B}} \in \{0, 1, \ldots, \delta^{-1} - 1\}^{d \times N}$, for each dimension $d$ and each token $N$, representing it in a $\delta^{-1}$ system with $Nd$ digits ($N$ tokens are ordered). Furthermore, *dummy indices* are needed to reduce the approximation error in the next step.

3. The network, with $\delta^{2(N+1)d} - 1$ loops, approximately maps *contextual token IDs* to the output embeddings of each token in a token-wise manner. *Contextual token IDs* refer to token IDs assigned to each token within the context of a sequence ID. To achieve a small approximation error, the network has to be designed so that neighboring IDs correspond to similar output token embeddings.

The details for each steps are provided below.

**Step 1. Token-wise Quantization.** The input space for each token $\boldsymbol{x} \in [0,1]^d$ are divided into $\delta$-discretized cubes, denoted by $\{Q_{\boldsymbol{\beta}}\}$ for $\boldsymbol{\beta} \in \{0, 1, \ldots, \delta^{-1} - 1\}^d$, defined as

$$Q_{\boldsymbol{\beta}} := \left\{ \boldsymbol{x} \in [0,1]^d : \boldsymbol{x}_i \in \left[\boldsymbol{\beta}_i \delta, \ \boldsymbol{\beta}_i \delta + 1\right), \quad i = 1, 2, \ldots, d \right\}.$$

By Lemma A.4, there exist a feed-forward layer $\mathrm{FF}^{(1)} : \mathbb{R}^{5d} \to \mathbb{R}^{5d}$ of width size $q = 7d$, and two affine linear maps $\mathcal{L}_1^{(1)} : \mathbb{R}^d \to \mathbb{R}^{5d}$ and $\mathcal{L}_2'^{(1)} : \mathbb{R}^{5d} \to \mathbb{R}^d$ such that

$$\mathcal{L}_2'^{(1)} \circ \left(\mathrm{id} + \mathrm{FF}^{(1)}\right)^{\circ(\delta^{-1} - 1)} \circ \mathcal{L}_1^{(1)}(\boldsymbol{x}) = \boldsymbol{\beta} \quad s.t. \quad \boldsymbol{x}_i \in \left[\boldsymbol{\beta}_i \delta, \ \boldsymbol{\beta}_i \delta + 1\right)$$

for any $i = 1, 2, \ldots, d$. Let $\mathcal{B}$ denote the output of the token-wise application to the input $\boldsymbol{x} \in [0,1]^{d \times N}$.

In addition, we need to bijectively map the $d$-dimensional vector $\boldsymbol{\beta}$ to an integer *token ID*, denoted by $z$. We use a $\delta^{-1}$-base system; define the vector $\boldsymbol{u}_{(\delta^{-1})} \in \mathbb{R}^d$ as

$$\boldsymbol{u}_{(\delta^{-1})} := (\delta^{-(d-1)}, \delta^{-(d-2)}, \ldots, \delta^{-1}, 1)^{\top},$$

and define $z$ as

$$z := \boldsymbol{u}_{(\delta^{-1})}^{\top} \boldsymbol{\beta} \in \{0, 1, \ldots, \delta^{-d} - 1\}.$$

To implement this, we define $\mathcal{L}_2^{(1)} : \mathbb{R}^{5d} \to \mathbb{R}^d$ with $\mathcal{L}_2'^{(1)} : \mathbb{R}^{5d} \to \mathbb{R}$ via

$$\mathcal{L}_2^{(1)}(\boldsymbol{x}) = \boldsymbol{u}_{(\delta^{-1})}^{\top} \mathcal{L}_2'^{(1)}(\boldsymbol{x}).$$

Thus, we have

$$\left( \boldsymbol{\mathcal{L}}_2^{(1)} \circ (\mathrm{FF}_1^{(1)})^{\circ(\delta^{-1} - 1)} \circ \boldsymbol{\mathcal{L}}_1^{(1)}(\boldsymbol{x}) \right)_n = \boldsymbol{u}_{(\delta^{-1})}^{\top} \boldsymbol{\beta} = z \quad s.t. \quad \boldsymbol{x}_{:,n} \in Q_{\boldsymbol{\beta}},$$

for $n = 1, 2, \ldots, N$.

We define the input cubes for each token assigned to $z$ as follows:

$$Q_z := \left\{ \boldsymbol{x} \in [0,1]^d : \boldsymbol{x}_i \in \left[\boldsymbol{\beta}_i \delta, \ \boldsymbol{\beta}_i \delta + 1\right) \quad \text{for } i = 1, 2, \ldots, d \quad s.t. \quad z = \boldsymbol{u}_{(\delta^{-1})}^{\top} \boldsymbol{\beta} \right\}.$$

We can confirm that

$$\|\boldsymbol{x}_z - \boldsymbol{x}_{z-1}\|_2 \leq \delta \sqrt{d} \quad \forall \boldsymbol{x}_z \in Q_z, \forall \boldsymbol{x}_{z-1} \in Q_{z-1} \quad s.t. \quad \boldsymbol{\beta}_d = 1, 2, \ldots, \delta^{-1} - 1.$$

While, we have

$$\|\boldsymbol{x}_z - \boldsymbol{x}_{z-1}\|_2 \leq \sqrt{d} \quad \forall \boldsymbol{x}_z \in Q_z, \forall \boldsymbol{x}_{z-1} \in Q_{z-1} \quad s.t. \quad \boldsymbol{\beta}_d = 0. \tag{11}$$

Informally, in this $\delta^{-1}$-base representation, the least significant digit corresponds to the index of the $d$-th dimension, $\boldsymbol{\beta}_d$. When incrementing the token ID one by one, the index in the $d$-th dimension increases sequentially $0, 1, 2, \ldots, \delta^{-1} - 1$, without changing the other digits; thus, consecutive IDs imply that the tokens are "similar" to each other in $d$-dimensional space, with a distance of at most $\delta\sqrt{d}$. However, when a carry occurs, the higher-order digits may change significantly, causing tokens that were not originally adjacent to appear next to each other in terms of their indices, as we are essentially projecting a $d$-dimensional space onto a 1-dimensional space. In this case, the distance is only bounded by $\sqrt{d}$.

**Step 2. Contextual Mapping.** The networks, with $N$-loops, map the list of $N$ token IDs, denoted by $\boldsymbol{z} \in \{0, 1, \ldots, \delta^{-d} - 1\}^N$, to sequence IDs bijectively. Furthermore, this mapping is not only bijective; it also requires the inclusion of additional *dummy indices*.

**Note:** We consider only the case where all $N$ input tokens are distinct, disregarding other cases, which can be treated as negligible when $\delta$ is small. The number of subsets where one of the $N$ tokens is duplicated is

$$\left(\delta^{-d}\right)^N - \delta^{-d} \cdot \left(\delta^{-d} - 1\right) \ldots \left(\delta^{-d} - N - 1\right) < C \delta^{-(N-1)d},$$

where $C$ is a constant. The volume of these subsets is $C\delta^{-(N-1)d} / \delta^{-Nd} = C\delta^d$, so the error with respect to the $L^p$ norm is $\mathcal{O}(\delta^d)$.

Let $\mathbb{L}_\delta$ denote the set, composed of distinct $N$ tokens, *i.e.*

$$\mathbb{L}_\delta := \{\boldsymbol{z} \in \{0, 1, \ldots, \delta^{-d} - 1\}^N \mid \boldsymbol{z}_i \neq \boldsymbol{z}_j \text{ for all } i \neq j\}.$$

Due to permutation equivalence, we can assume without loss of generality that elements of $\boldsymbol{z} \in \mathbb{L}_\delta$ is ordered, *i.e.*, $\boldsymbol{z}_1 > \boldsymbol{z}_2 > \cdots > \boldsymbol{z}_N$. Define $\boldsymbol{u}_{(\delta^{-d})} := (\delta^{-(N-1)d}, \ldots, \delta^{-d}, 1)^\top$, which satisfy

$$\left\| \boldsymbol{u}_{(\delta^{-d})}^\top \boldsymbol{z} - \boldsymbol{u}_{(\delta^{-d})}^\top \boldsymbol{z}' \right\| > 1, \quad \text{if } \boldsymbol{z} \neq \boldsymbol{z}' \text{ for any } \boldsymbol{z}, \boldsymbol{z}' \in \mathbb{L}_\delta.$$

This mapping, $\boldsymbol{u}_{(\delta^{-d})}^\top \boldsymbol{z}$, represents $\boldsymbol{z}$ in a $\delta^{-d}$-base system. Then, we define sequence ID of $\boldsymbol{z} \in \mathbb{L}_\delta$ as:

$$\mathrm{s}(\boldsymbol{z}) := \boldsymbol{u}_{(\delta^{-d})}^\top \boldsymbol{z} = \sum_{n=1}^{N} \boldsymbol{z}_n \delta^{-(N-n)d}. \tag{12}$$

By Lemma A.5, there exist a Transformer block $\mathrm{TF}'^{(2)} : \mathbb{R}^{3 \times N} \to \mathbb{R}^{3 \times N}$ with single head, head size $s = 1$, and width size $q = 3$, and two affine linear maps $\mathcal{L}_1'^{(2)} : \mathbb{R} \to \mathbb{R}^3$ and $\mathcal{L}_2'^{(2)} : \mathbb{R}^3 \to \mathbb{R}$ such that

$$\mathcal{L}_2'^{(2)} \circ \mathrm{TF}'^{(2) \circ N} \circ \mathcal{L}_1'^{(2)}(\boldsymbol{z}^\top) = \mathrm{s}(\boldsymbol{z}) \cdot \mathbf{1}_N^\top,$$

where $\boldsymbol{z}^\top \to \boldsymbol{u}_{(\delta^{-d})}^\top \boldsymbol{z}$ is a contextual mapping.

Furthermore, we have to add *dummy indices* to alleviate the approximation error caused by the looped architecture in Step 3. Recall that $\boldsymbol{\mathcal{B}} \in \{0, 1, \ldots, \delta^{-1} - 1^{d \times N}\}$ represents the coordinates of the inputs. Let $\boldsymbol{Z} \in \{0, 1, \ldots, \delta^{-1} - 1\}^{d \times N}$ denote the coordinates where the tokens are ordered by their token IDs, i.e., $\boldsymbol{u}_{(\delta^{-1})}^\top \boldsymbol{Z}_1 > \boldsymbol{u}_{(\delta^{-1})}^\top \boldsymbol{Z}_2 > \cdots > \boldsymbol{u}_{(\delta^{-1})}^\top \boldsymbol{Z}_N$, *i.e.*, $\boldsymbol{z} = \boldsymbol{u}_{(\delta^{-1})}^\top \boldsymbol{Z}$. Thus, by redefining the sequence ID of Eq. 12 for $\boldsymbol{Z}$ instead of $\boldsymbol{z}$, sequence IDs in $\delta^{-d}$-base can be rewritten in the $\delta^{-1}$-base system as follows:

$$\mathrm{s}(\boldsymbol{Z}) = \boldsymbol{u}_{(\delta^{-d})}^\top (\boldsymbol{u}_{(\delta^{-1})}^\top \boldsymbol{Z})$$
$$= \sum_{i=1}^{d} \sum_{n=1}^{N} \boldsymbol{Z}_{i,n} \delta^{-\left((N-n)d + (d-i)\right)}.$$

We also define the sequence IDs for each $\boldsymbol{\mathcal{B}}$ as:

$$\mathrm{s}(\boldsymbol{\mathcal{B}}) = \sum_{i=1}^{d} \sum_{n=1}^{N} \boldsymbol{\mathcal{B}}_{i,n} \delta^{-\left((N-n)d + (d-i)\right)}.$$

Then, we define *extended* sequence IDs as:

$$\mathrm{s}_{\mathrm{extend}}(\boldsymbol{Z}) := 2\mathrm{s}(\boldsymbol{Z}) - \boldsymbol{Z}_{d,N}$$
$$= \boldsymbol{Z}_{d,N} + \sum_{\substack{i=1 \\ (i,n) \neq (d,N)}}^{d} \sum_{n=1}^{N} 2\boldsymbol{Z}_{i,n} \delta^{-\left((N-n)d + (d-i)\right)},$$

$$\mathrm{s}_{\mathrm{extend}}(\boldsymbol{\mathcal{B}}) := \boldsymbol{\mathcal{B}}_{d,N} + \sum_{\substack{i=1 \\ (i,n) \neq (d,N)}}^{d} \sum_{n=1}^{N} 2\boldsymbol{\mathcal{B}}_{i,n} \delta^{-\left((N-n)d + (d-i)\right)}.$$

Recall that $\boldsymbol{\mathcal{B}}_{d,N} \in \{0, 1, \ldots, \delta^{-1} - 1\}$. We define the *dummy indices* as:

$$\mathrm{s}_{\mathrm{dummy}}^b(\boldsymbol{\mathcal{B}}) := b + \sum_{\substack{i=1 \\ (i,n) \neq (d,N)}}^{d} \sum_{n=1}^{N} 2\boldsymbol{\mathcal{B}}_{i,n} \delta^{-\left((N-n)d + (d-i)\right)}, \quad \text{for } b = \delta^{-1}, \delta^{-1} + 1, \ldots, 2\delta^{-1} - 1.$$

We define each set as follows:

$$\mathcal{S}_{\mathrm{distinct}} := \{\mathrm{s}_{\mathrm{extend}}(\boldsymbol{Z})\}, \quad \mathcal{S}_{\mathrm{distinct+duplicate}} := \{\mathrm{s}_{\mathrm{extend}}(\boldsymbol{\mathcal{B}})\},$$
$$\mathcal{S}_{\mathrm{dummy}} := \{\mathrm{s}_{\mathrm{dummy}}^b(\boldsymbol{\mathcal{B}}) \mid b = \delta^{-1}, \delta^{-1} + 1, \ldots, 2\delta^{-1} - 1\}.$$

Then, we can see that

$$\mathcal{S}_{\text{distinct}} \subset \mathcal{S}_{\text{distinct+duplicate}}, \quad \mathcal{S}_{\text{distinct+duplicate}} \cap \mathcal{S}_{\text{dummy}} = \emptyset,$$
$$\mathcal{S}_{\text{distinct+duplicate}} \cup \mathcal{S}_{\text{dummy}} = \{0, 1, \ldots, 2\delta^{-Nd} - 1\}. \tag{13}$$

We define the input cubes for each token assigned to extended sequence ID $s$ as follows:

$$\boldsymbol{Q}_s := \left\{ \mathbf{X} \in [0,1]^{d \times N} : \mathbf{X}_{i,n} \in \left[\boldsymbol{\mathcal{B}}_{i,n}\delta, \boldsymbol{\mathcal{B}}_{i,n}\delta + 1\right) \quad \text{for } i = 1, 2, \ldots, Nd \quad s.t. \quad s = \mathrm{s}_{\text{extend}}(\boldsymbol{\mathcal{B}}) \right\}.$$

We can confirm for that

$$\|\boldsymbol{X}_s - \boldsymbol{X}_{s-1}\|_2 \leq \delta\sqrt{Nd} \quad \forall \boldsymbol{X}_s \in Q_s, \forall \boldsymbol{X}_{s-1} \in Q_{s-1} \quad s.t. \quad \boldsymbol{\mathcal{B}}_{d,N} = 1, 2, \ldots, \delta^{-1} - 1. \tag{14}$$

While, we have

$$\|\boldsymbol{X}_s - \boldsymbol{X}_{s-1}\|_2 \leq \sqrt{Nd} \quad \forall \boldsymbol{X}_s \in Q_s, \forall \boldsymbol{X}_{s-1} \in Q_{s-1} \quad s.t. \quad \boldsymbol{\mathcal{B}}_{d,N} = 0. \tag{15}$$

Informally, this set represents a collection of consecutive sequence IDs, where the first $N-1$ tokens are identical, and the $N$-th token is "similar". For example: (1) I write 'a paper', and (2) I write 'papers'. As explained in token IDs, if only the $d$-th dimension of the $N$-th token differs, then the tokens remain adjacent in the $d$-dimensional space. As a result, the total variation in the input sequence within the set is bounded by $\delta\sqrt{d}$.

To implement this, we slightly modified $\text{TF}'_2$. By Corollary A.6, there exist a Transformer block $\text{TF}^{(2)} : \mathbb{R}^{5 \times N} \to \mathbb{R}^{5 \times N}$ with single head, head size $s = 1$, and width size $q = 4$, and two affine linear maps $\mathcal{L}_1^{(2)} : \mathbb{R}^2 \to \mathbb{R}^5$ and $\mathcal{L}_2^{(2)} : \mathbb{R}^5 \to \mathbb{R}$ such that

$$\boldsymbol{\mathcal{L}}_2^{(2)} \circ \text{TF}^{(2)\circ N} \circ \boldsymbol{\mathcal{L}}_1^{(2)}\left(\begin{bmatrix} \boldsymbol{z}^\top \\ \boldsymbol{Z}_{d,:} \end{bmatrix}\right) = \mathrm{s}_{\text{extend}}(\boldsymbol{z}) \cdot \mathbf{1}_N^\top \quad \text{for any } \boldsymbol{z} \in \mathbb{L}_\delta.$$

**Step 3. Token-wise Mapping.** From Step 1 and Step 2, the network takes a token ID and sequence ID as input for each token, which together form a *contextual token ID*. The network with $\delta^{-2(N+1)d} - 1$ loops approximately maps these *contextual token IDs* to the output token embeddings of the target function.

To construct contextual token IDs, we define a bijective mapping $\mathcal{L}_0^{(3)} : \mathbb{N}^2 \to \mathbb{N}$ as follows:

$$\mathcal{L}_0^{(3)}(z, s) := 2\delta^{-Nd}z + s,$$

where $z$ represents a token ID and $s$ represents a extended sequence ID. Note that sequence IDs are less than $2\delta^{-Nd}$, so informally, it's as if we are adding another digit, $\boldsymbol{z}_n$, as the most significant digit in a $\delta^{-d}$-based system. Define the set of contextual token IDs for distinct $N$ token IDs as:

$$\mathcal{K}_{\text{distinct}} := \left\{ \mathcal{L}_3\left(\boldsymbol{z}_n, \mathrm{s}_{\text{extend}}(\boldsymbol{Z})\right) : n = 1, 2, \ldots, N \right\}.$$

We also define the set for all inputs, including cases where some token IDs are duplicated, as

$$\mathcal{K}_{\text{distinct+duplicate}} := \left\{ \mathcal{L}_3\left(\boldsymbol{\beta}_n, \mathrm{s}_{\text{extend}}(\boldsymbol{\mathcal{B}})\right) : n = 1, 2, \ldots, N \right\}$$
$$= \left\{ s + 2\delta^{-Nd}z : s \in \mathcal{S}_{\text{distinct+duplicate}}, z = 0, 1, \cdots, \delta^{-d} \right\}$$
$$= \left\{ \boldsymbol{\mathcal{B}}_{d,N} + \sum_{\substack{i=1 \\ (i,n) \neq (d,N)}}^{d} \sum_{n=1}^{N} 2\boldsymbol{\mathcal{B}}_{i,n}\delta^{-\left((N-n)d+(d-i)\right)} + 2\delta^{-Nd}z \right\},$$

and dummy contextual token ID as

$$\mathcal{K}_{\text{dummy}} := \left\{ s + 2\delta^{-Nd}z : s \in \mathcal{S}_{\text{dummy}}, z = 0, 1, \cdots, \delta^{-d} \right\}.$$

We can confirm, from Eq. 13,

$$\mathcal{K}_{\text{distinct}} \subset \mathcal{K}_{\text{distinct+duplicate}}, \quad \mathcal{K}_{\text{distinct+duplicate}} \cap \mathcal{K}_{\text{dummy}} = \emptyset,$$

$$\mathcal{K}_{\text{distinct+duplicate}} \cup \mathcal{K}_{\text{dummy}} = \{0, 1, \dots, 2\delta^{-(N+1)d} - 1\}.$$

We now define the target output token embedding for each extended contextual mapping ID. Define $g : \mathcal{K} \to \mathbb{R}^d$, the mapping from extended contextual IDs to their token embeddings, as follows:

$$g(k) = \begin{cases} \bar{f}(\boldsymbol{X})_{:,n} \quad s.t. \quad \mathcal{L}_3\big(\boldsymbol{\beta}_n, s_{\text{extend}}(\boldsymbol{\mathcal{B}})\big) = k & \text{for } k \in \mathcal{K}_{\text{distinct+duplicate}}, \\ \text{lin\_interp}\big(g(\text{near}^-(k)), g(\text{near}^+(k)), k - \text{near}^-(k), \delta^{-1}\big) & \text{for } k \in \mathcal{K}_{\text{dummy}}. \end{cases}$$

When $\boldsymbol{\mathcal{B}}_{d,n} \neq 0$, from Eq. 14, we have

$$g(k) - g(k-1) \leq \omega_f^{cont}(\delta\sqrt{d}),$$

since the token $\boldsymbol{z}_n$ remains the same for both $k$ and $k-1$. While $\boldsymbol{\mathcal{B}}_{d,n} = 0$, a carryover occurs, and we only bound the input sequences as in Eq.15, with the possibility that the token ID differs, the variation is bounded as Eq.11. Since we have dummy token IDs, there are redundant indices before the carryover occurs. As a result, the variation in the next index is bounded by a factor of $\delta$, *i.e.*,

$$\|g(k) - g(k-1)\|_p \leq \delta\big(\omega_f^{cont}(\sqrt{Nd}) + \omega_f^{tok}(\sqrt{d})\big).$$

Since that $\omega_f^{cont,tok}(n \cdot t) \leq n \cdot \omega_f^{cont,tok}(t)$ for any $n \in \mathbb{N}$ and $t \in [0, \infty)$ with $\delta < 1$, we have

$$\|g(k) - g(k-1)\|_p \leq \omega_f^{tok}(\delta\sqrt{d}) + \omega_f^{cont}(\delta\sqrt{Nd}).$$

Thus, we have

$$|(g(k)_i - g(k-1)_i| \leq \omega_f^{tok}(\delta\sqrt{d}) + \omega_f^{cont}(\delta\sqrt{Nd}) \quad \text{for } i = 1, 2, \dots, d.$$

By Lemma 4.1, there exist feed-forward layer $\text{FF}^{(3)} : \mathbb{R}^{14d} \to \mathbb{R}^{14d}$ of width size $20d$ and two affine linear maps $\mathcal{L}_1^{(3)} : \mathbb{R}^d \to \mathbb{R}^{14d}$ and $\mathcal{L}_2^{(3)} : \mathbb{R}^{14d} \to \mathbb{R}^d$ such that

$$\big|\big(\mathcal{L}_2^{(3)} \circ (\text{id} + \text{FF}^{(3)})^{(2\delta^{-(N+1)d}-1)} \circ \mathcal{L}_1^{(3)}(k) - g(k)\big)_i\big| \leq \omega_f^{tok}(\delta\sqrt{d}) + \omega_f^{cont}(\delta\sqrt{Nd}),$$

for any $i = 1, 2, \dots, d$ and $k = 0, 1, \dots, 2\delta^{-(N+1)d} - 1$.

**Consolidation into Single Looped Transformer.** Lastly, we demonstrate $\tilde{f}$ can be represented in Looped Transformers. Let

$$\boldsymbol{X}^{(0)} \in \mathbb{R}^{1 \times N}, \quad \boldsymbol{X}^{(1)} \in \mathbb{R}^{5d \times N}, \quad \boldsymbol{X}^{(2)} \in \mathbb{R}^{5d \times N}, \quad \boldsymbol{X}^{(3)} \in \mathbb{R}^{14d \times N}$$

denote the divided input space. Define single head $\text{Attn} : \mathbb{R}^{(24d+1) \times N} \to \mathbb{R}^{(24d+1) \times N}$ with head size 1 as:

$$\text{Attn}\left(\begin{bmatrix} \boldsymbol{X}^{(0)} \\ \boldsymbol{X}^{(1)} \\ \boldsymbol{X}^{(2)} \\ \boldsymbol{X}^{(3)} \end{bmatrix}\right) = \begin{bmatrix} \boldsymbol{0}_{5d+1 \times N} \\ \text{Attn}^{(2)}(\boldsymbol{X}^{(2)}) \\ \boldsymbol{0}_{14d \times N} \end{bmatrix},$$

where $\text{Attn}^{(2)}$ denote the self-attention layer of $\text{TF}^{(2)}$. Let

$$\boldsymbol{x}_0 \in \mathbb{R}, \quad \boldsymbol{x}_1 \in \mathbb{R}^{5d}, \quad \boldsymbol{x}_2 \in \mathbb{R}^{5d}, \quad \boldsymbol{x}_3 \in \mathbb{R}^{14d}$$

denote the token-wise input space. Define $\text{FF} : \mathbb{R}^{24d+1} \to \mathbb{R}^{24d+1}$, with impulse function defined in Proposition A.3, as:

$$\text{FF}\left(\begin{bmatrix} \boldsymbol{x}_0 \\ \boldsymbol{x}_1 \\ \boldsymbol{x}_2 \\ \boldsymbol{x}_3 \end{bmatrix}\right) = \left(\begin{bmatrix} \boldsymbol{x}_0 + 1 \\ \text{FF}^{(1)}(\boldsymbol{x}_1) \\ \text{FF}^{(2)}(\boldsymbol{x}_2) + \textbf{impulse}_{(\delta^{-1}-1)}\big(\mathcal{L}_1^{(2)} \circ \mathcal{L}_2^{(1)}(\boldsymbol{x}_1, (\boldsymbol{x}_1)_d)\big) \\ \text{FF}^{(3)}(\boldsymbol{x}_3) + \textbf{impulse}_{(\delta^{-1}+N)}\big(\mathcal{L}_1^{(3)} \circ \mathcal{L}_0^{(3)}(\boldsymbol{x}_1, \boldsymbol{x}_2)\big) \end{bmatrix}\right)$$

where $\mathrm{FF}^{(2)}$ denotes the feed-forward layer of $\mathrm{TF}^{(2)}$, and **impulse** refers to the dimension-wise application of impulse. Note that $\boldsymbol{x}_0$ serves the role of a counter. As shown in Proposition A.3, the impulse requires 4 ReLU functions per dimension. With $19d$ dimensions, this results in an additional width of $72d+4$. The total width consists of $7d$ for $\mathrm{FF}^{(1)}$, 4 for $\mathrm{TF}^{(2)}$, and $20d$ for $\mathrm{FF}^{(3)}$, resulting in a total width of $99d+8$. Define two affine linear maps $\mathcal{L}_1 : \mathbb{R}^d \to \mathbb{R}^{24d+1}$ and $\mathcal{L}_2 : \mathbb{R}^{24d+1} \to \mathbb{R}^d$ such that

$$\mathcal{L}_1(\boldsymbol{x}) = (0, \mathcal{L}_1^{(1)}(\boldsymbol{x}), \boldsymbol{0}_{19d})^\top, \quad \mathcal{L}_2((\boldsymbol{x}_0, \boldsymbol{x}_1, \boldsymbol{x}_2, \boldsymbol{x}_3)^\top) = \mathcal{L}_2^{(3)}(\boldsymbol{x}_3).$$

Thus, we have

$$\tilde{f}(\boldsymbol{X}) = \mathcal{L}_2 \circ \mathrm{TF}^{\circ(\delta^{-1}+N+2\delta^{-(N+1)d})} \circ \mathcal{L}_1(\boldsymbol{X})$$

$$s.t. \quad \|\tilde{f}(\boldsymbol{X}) - \bar{f}(\boldsymbol{X})\|_p \le \omega_f^{tok}(\delta\sqrt{d}) + \omega_f^{cont}(\delta\sqrt{Nd}) + \mathcal{O}(\delta^d).$$

**Deriving Approximation Error.** Generally, the following inequality holds.

$$\sum_{i=1}^n x_i^p \le \left(\sum_{i=1}^n x_i\right)^p \quad \text{for } x_i \ge 0 \text{ and } p \ge 1.$$

Substituting $x_i = \|\bar{f}(\boldsymbol{X}) - f(\boldsymbol{X})\|_p$ into the above inequality results in

$$\|\bar{f} - f\|_{L^p([0,1]^{d \times N})} = \left(\int \|\bar{f}(\boldsymbol{X}) - f(\boldsymbol{X})\|_p^p \, d\boldsymbol{X}\right)^{1/p} \le \int \|\bar{f}(\boldsymbol{X}) - f(\boldsymbol{X})\|_p d\boldsymbol{X}. \quad (16)$$

Also we can bound the entire norm with the token-wise norm as:

$$\|\tilde{f}(\boldsymbol{X}_k) - \bar{f}(\boldsymbol{X}_k)\|_p \le \max_n \|\tilde{f}(\boldsymbol{X}_k)_{:,n} - \bar{f}(\boldsymbol{X}_k)_{:,n}\|_p \le \max_{n,k' \in \mathcal{K}} \|\bar{f}(\boldsymbol{X}_{k'})_{:,n} - \bar{f}(\boldsymbol{X}_{k'-1})_{:,n}\|_p.$$

With the triangle inequality, we have approximation error as

$$\begin{aligned}
\|\tilde{f} - f\|_{L^p([0,1]^{d \times N})} &\le \|\tilde{f}(\boldsymbol{X}) - f(\boldsymbol{X})\|_p \cdot 1 \\
&\le \|\tilde{f}(\boldsymbol{X}) - \bar{f}(\boldsymbol{X})\|_p + \|\bar{f}(\boldsymbol{X}) - f(\boldsymbol{X})\|_p \\
&\le \max_{n,k' \in \mathcal{K}} \|\bar{f}(\boldsymbol{X}_{k'})_{:,n} - \bar{f}(\boldsymbol{X}_{k'-1})_{:,n}\|_p + \|\bar{f}(\boldsymbol{X}) - f(\boldsymbol{X})\|_p + \mathcal{O}(\delta^d) \\
&\le \omega_f^{tok}(\delta\sqrt{d}) + \omega_f^{cont}(\delta\sqrt{Nd}) + \omega_f(\delta\sqrt{Nd}) + \mathcal{O}(\delta^d).
\end{aligned}$$

Then, $\delta$ must be expressed in terms of the number of loops $r$ to determine the approximation rate.

$$\begin{aligned}
r = \delta^{-1} + N + 2\delta^{-(N+1)d} &\Leftrightarrow \delta^{-1} + 2\delta^{-(N+1)d} = r - N \\
&\Leftrightarrow \delta^{-1} \cdot 2\delta^{-(N+1)d} \ge r - N \\
&\Leftrightarrow 2\delta^{-(N+1)d-1} \ge r - N \\
&\Leftrightarrow \delta \le \left(\frac{r-N}{2}\right)^{-1/\left((N+1)d+1\right)}. \quad (17)
\end{aligned}$$

Thus, we can derive Theorem 3.7. $\qquad\square$

## A.3 PIECEWISE LINEAR FUNCTIONS

Here, we define three functions implemented with ReLU functions.

**Proposition A.1** (Rectangular function). *Given $t \in \mathbb{R}$, define $\mathrm{rect}_t : \mathbb{R} \to \mathbb{R}$ as:*

$$\mathrm{rect}_t(x) = \begin{cases} 1 & \text{if } x \in [t, t+1), \\ 0 & \text{otherwise}. \end{cases}$$

*Four ReLU functions can approximate $\mathrm{rect}_t$ with arbitrarily small error via*

$$\phi_t(x) := \sigma_R\left(\tfrac{x-t+\epsilon}{\epsilon}\right) - \sigma_R\left(\tfrac{x-t}{\epsilon}\right) + \sigma_R\left(\tfrac{-x+t+1}{\epsilon}\right) - \sigma_R\left(\tfrac{-x+t+1-\epsilon}{\epsilon}\right) - 1,$$

*where $\lim_{\epsilon \to 0} \phi_t(x) = \mathrm{rect}_t(x)$.*

**Proposition A.2** (Step function). *Define* $\text{step} : \mathbb{R} \to \mathbb{R}$ *as:*

$$\text{step}(x) = \begin{cases} 0 & \text{if } x < 0, \\ 1 & \text{if } x \geq 0. \end{cases}$$

*Two* ReLU *functions can approximate* step *with arbitrarily small error via*

$$\phi(x) := \sigma_R\big(\tfrac{x}{\epsilon} + 1\big) - \sigma_R\big(\tfrac{x}{\epsilon}\big),$$

*where* $\lim_{\epsilon \to 0} \phi(x) = \text{step}(x)$.

**Proposition A.3** (Impulse function). *Given* $\theta \in \mathbb{N}$, *define* $\text{impulse}_\theta : \mathbb{R} \to \mathbb{R}$ *for* $x \in [-M, M]$ *and* $t \in \mathbb{N}$ *as:*

$$\text{impulse}_\theta(x, t) = \begin{cases} x & \text{if } t = \theta, \\ 0 & \text{otherwise.} \end{cases}$$

*Four* ReLU *functions can approximate* $\text{impulse}_\theta$ *with arbitrarily small error via*

$$\text{impulse}_\theta(x, t) := \sigma_R\big(x + 2M(t - \theta + 1/2)\big) - 2M\sigma_R(t - \theta + 1/2)$$
$$- \sigma_R\big(x + 2M(t - \theta - 1/2)\big) + 2M\sigma_R(t - \theta - 1/2).$$

### A.4 STEP 1. TOKEN-WISE QUANTIZATION

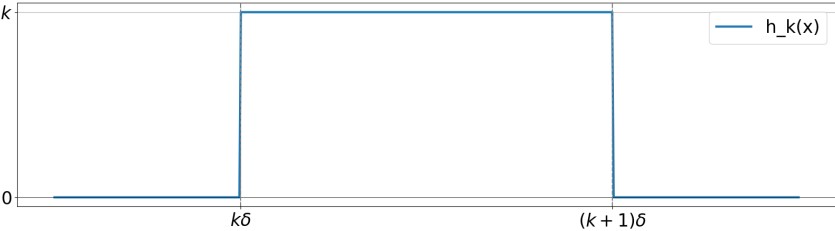

Figure 4: An illustration of $h_k(x)$.

We aim to construct quantization function $\mathbf{g} : [0, 1]^d \to \{0, 1, \dots, \delta^{-1}\}^d$ for each dimension as

$$\mathbf{g}(\boldsymbol{x}) = (g(\boldsymbol{x}_1), g(\boldsymbol{x}_2), \dots, g(\boldsymbol{x}_d))^\top,$$
$$\text{where} \quad g(x) = \begin{cases} k & \text{if } x \in [k\delta, (k+1)\delta), \\ 0 & \text{otherwise.} \end{cases}$$

This function $g : \mathbb{R} \to \mathbb{R}$ can be expressed as

$$g(x) = \sum_{i=0}^{n-1} i \cdot \text{rect}_i(x)$$

for any $n \in \mathbb{N}$ and $x \in \mathbb{R}$. The illustration of $h_k(x) := k \cdot \text{rect}_k(x)$ is shown in Fig 4. The key idea is that $h_k(x)$ can be represented with a single function $h$ in the form of $h\big(kx, k^2, k\big)$. Lemma A.4 implement $h\big(kx, k^2, k\big)$ with a feed-forward layer and perform the summation through a skip connection.

**Lemma A.4.** *Given any* $\delta^{-1} \in \mathbb{N}$ *and* $\boldsymbol{x} \in \mathbb{R}^d$, *there exist a feed-forward layer* $\text{FF} : \mathbb{R}^{5d} \to \mathbb{R}^{5d}$ *of width size* $q = 7d$, *and two affine linear maps* $\mathcal{L}_1 : \mathbb{R}^d \to \mathbb{R}^{5d}$ *and* $\mathcal{L}_2 : \mathbb{R}^{5d} \to \mathbb{R}^d$ *such that*

$$\left(\mathcal{L}_2 \circ \big(\text{id} + \text{FF}\big)^{\circ(\delta^{-1}-1)} \circ \mathcal{L}_1(\boldsymbol{x})\right)_i = \begin{cases} k & \text{if } \boldsymbol{x}_i \in [k\delta, (k+1)\delta), \ k = 0, \dots, \delta^{-1} - 1, \\ 0 & \text{otherwise,} \end{cases}$$

*for any* $i = 1, 2, \dots, d$.

*Proof.* On the basis of proposition A.1, define function $h_k(x) = k \cdot \text{rect}_k(x)$ via

$$h_k(x) := \sigma_R\big(\tfrac{k}{\epsilon}(x - \tfrac{k}{\delta} + \epsilon)\big) - \sigma_R\big(\tfrac{k}{\epsilon}(x - \tfrac{k}{\delta})\big) + \sigma_R\big(\tfrac{k}{\epsilon}(-x + \tfrac{k}{\delta} + 1)\big)$$
$$- \sigma_R\big(\tfrac{k}{\epsilon}(-x + \tfrac{k}{\delta} + 1 - \epsilon)\big) - k,$$

which satisfies

$$h_k(x) = \begin{cases} k & \text{if } x \in [k\delta, k\delta + 1), \\ 0 & \text{otherwise.} \end{cases}$$

For any $x \in [k\delta, k\delta + 1)$ where $k = 0, 1, \ldots, \delta^{-1} - 1$, we have

$$\sum_{i=0}^{\delta^{-1}-1} h_i(x) = h_k(x) = k.$$

Define function $h : \mathbb{R}^3 \to \mathbb{R}$ to represent $h_k$ via

$$h(kx, k^2, k) := \sigma_R\left(\frac{kx}{\epsilon} - \frac{k^2}{\epsilon} + k\right) - \sigma_R\left(\frac{kx}{\epsilon} - \frac{k^2}{\epsilon}\right) + \sigma_R\left(-\frac{kx}{\epsilon} + \frac{k^2}{\epsilon} + \frac{k}{\epsilon}\right)$$
$$- \sigma_R\left(-\frac{kx}{\epsilon} + \frac{k^2}{\epsilon} + k\frac{1-\epsilon}{\epsilon}\right) - \sigma_R(k) = h_k(x).$$

Define $\boldsymbol{\xi}_k$ as

$$\boldsymbol{\xi}_k = \left(kx, \ k^2, \ k, \ x, \ \sum_{i=0}^{k-1} h_i(x)\right)^\top.$$

Then, construct a feed-forward layer $\text{FF} : \mathbb{R}^5 \to \mathbb{R}^5$ with a skip connection such that

$$\big(\text{id} + \text{FF}\big)(\boldsymbol{\xi}_k) = \big(\text{id} + \text{FF}\big)\left(\Big(kx, \ k^2, \ k, \ x, \ \sum_{i=0}^{k-1} h_i(x)\Big)^\top\right)$$

$$= \left(\Big((k+1)x, \ (k+1)^2, \ k+1, \ x, \ \sum_{i=0}^{k} h_i(x)\Big)\right)^\top$$

$$= \boldsymbol{\xi}_{k+1}.$$

via

$$\big(\text{id} + \text{FF}\big)\left(\begin{bmatrix} kx \\ k^2 \\ k \\ x \\ \sum_{i=0}^{k-1} h_i(x) \end{bmatrix}\right) = \begin{bmatrix} kx \\ k^2 \\ k \\ x \\ \sum_{i=0}^{k-1} h_i(x) \end{bmatrix} +$$

$$\begin{bmatrix} 1 & 0 & 0 & 0 & 0 & 0 & 0 \\ 0 & 1 & 0 & 0 & 0 & 0 & 0 \\ 0 & 0 & 0 & 0 & 0 & 0 & 0 \\ 0 & 0 & 0 & 0 & 0 & 0 & 0 \\ 0 & 0 & 1 & -1 & 1 & -1 & -1 \end{bmatrix} \sigma_R\left(\begin{bmatrix} 0 & 0 & 0 & 1 & 0 \\ 0 & 0 & 2 & 0 & 0 \\ \frac{1}{\epsilon} & -\frac{1}{\epsilon} & 1 & 0 & 0 \\ \frac{1}{\epsilon} & -\frac{1}{\epsilon} & 0 & 0 & 0 \\ -\frac{1}{\epsilon} & \frac{1}{\epsilon} & 1 & 0 & 0 \\ -\frac{1}{\epsilon} & \frac{1}{\epsilon} & \frac{1-\epsilon}{\epsilon} & 0 & 0 \\ 0 & 0 & 1 & 0 & 0 \end{bmatrix}\begin{bmatrix} kx \\ k^2 \\ k \\ x \\ \sum_{i=0}^{k-1} h_i(x) \end{bmatrix} + \begin{bmatrix} 0 \\ 1 \\ 1 \\ 0 \\ 0 \end{bmatrix}\right)$$

$$= \begin{bmatrix} kx \\ k^2 \\ k \\ x \\ \sum_{i=0}^{k-1} h_i(x) \end{bmatrix} + \begin{bmatrix} x \\ 2k+1 \\ 1 \\ 0 \\ h_k(x) \end{bmatrix}$$

$$= \begin{bmatrix} kx + x \\ k^2 + 2k + 1 \\ k + 1 \\ x \\ \sum_{i=0}^{k-1} h_i(x) + h_k(x) \end{bmatrix}$$

$$= \begin{bmatrix} (k+1)x \\ (k+1)^2 \\ k + 1 \\ x \\ \sum_{i=0}^{k} h_i(x) \end{bmatrix}.$$

Then, define two affine linear maps $\mathcal{L}_1 : \mathbb{R}^1 \to \mathbb{R}^5$ and $\mathcal{L}_2 : \mathbb{R}^5 \to \mathbb{R}^1$ by

$$\mathcal{L}_1(x) := (0, 0, 0, x, 0), \quad \mathcal{L}_2(x_1, x_2, x_3, x_4, x_5) := x_5.$$

Thus, we have

$$
\begin{aligned}
\mathcal{L}_2 \circ (\mathrm{id} + \mathrm{FF})^{\circ(\delta^{-1}-1)} \circ \mathcal{L}_1(x) &= \mathcal{L}_2 \circ (\mathrm{id} + \mathrm{FF})^{\circ(\delta^{-1}-1)}(\boldsymbol{\xi}_1) \\
&= \mathcal{L}_2(\boldsymbol{\xi}_{\delta^{-1}}) \\
&= \sum_{i=0}^{\delta^{-1}-1} h_i(x).
\end{aligned}
$$

For $d$-dimensional inputs, we need $d$-times parameters. $\qquad\square$

## A.5    STEP 2. CONTEXTUAL MAPPING

The network takes token IDs as inputs, denoted by $\boldsymbol{z} \in \{0, 1, \dots, \delta^{-d} - 1\}^N$. We consider only the case where all token IDs are distinct, since this accounts for the majority when $\delta$ is small. The network maps token IDs into a sequence ID using inner product with the vector $\boldsymbol{u} \in \mathbb{R}^N$ defined as $\boldsymbol{u} := (\delta^{-(N-1)d}, \delta^{-(N-2)d}, \dots, \delta^{-d}, 1)^\top$ *i.e.*

$$\mathrm{CM}(\boldsymbol{z}) := \boldsymbol{u}^\top \boldsymbol{z}.$$

Due to permutation equivalence, we can assume without loss of generality that elements of $\boldsymbol{z} \in \mathbb{L}_\delta$ is ordered, *i.e.*, $\boldsymbol{z}_1 > \boldsymbol{z}_2 > \cdots > \boldsymbol{z}_N$. Then the map CM satisfies

$$\left\| \boldsymbol{u}^\top \boldsymbol{z} - \boldsymbol{u}^\top \boldsymbol{z}' \right\| > 1, \quad \text{if } \boldsymbol{z} \neq \boldsymbol{z}'.$$

In other words, CM represent $\boldsymbol{z}$ in $\delta^{-d}$-base system. The network computes $\boldsymbol{u}^\top \boldsymbol{z}$ in the form of $\sum_{i=1}^N \delta^{-(N-i)d} \boldsymbol{z}_i$. The network computes $\boldsymbol{s}^{(k)} := \sum_{i=1}^k \delta^{-(k-i)d} \boldsymbol{z}_i$ in each loop, and after $N$-loops, it outputs $\boldsymbol{s}^{(N)} = \boldsymbol{u}^\top \boldsymbol{z}$. To implement this, the self-attention layer select $\boldsymbol{z}_k$ in the $k$-th loop iteration. We design the key and value weights to select the maximum token ID. The feed-forward layer post-processes the token ID as if selected, then replaces it with negative value to prevent selection in subsequent iterations, *i.e.*, the post-processed token IDs for the $k$-th loop is

$$\boldsymbol{z}_i^{(k)} = z \quad s.t. \quad \begin{cases} z < 0 & \text{if } i \leq k, \\ z = \boldsymbol{z}_i & \text{otherwise.} \end{cases}$$

**Lemma A.5.** *Consider the set, composed of distinct indices for $d$-dimension $\delta$-discretized regions of $N$ tokens, i.e.*

$$\mathbb{L}_\delta := \{ \boldsymbol{z} \in \{0, 1, \dots, \delta^{-d} - 1\}^N \mid \boldsymbol{z}_i \neq \boldsymbol{z}_j \text{ for all } i \neq j \}.$$

*There exists a function $\mathrm{CM} : \mathbb{R}^N \to \mathbb{R}$ composed of Transformer block $\mathrm{TF} : \mathbb{R}^{3 \times N} \to \mathbb{R}^{3 \times N}$ with single head, head size $s = 1$, and width size $q = 3$, and two affine linear maps $\mathcal{L}_1 : \mathbb{R} \to \mathbb{R}^3$ and $\mathcal{L}_2 : \mathbb{R}^3 \to \mathbb{R}$, such that*

$$\boldsymbol{\mathcal{L}}_2 \circ \mathrm{TF}^{\circ N} \circ \boldsymbol{\mathcal{L}}_1(\boldsymbol{z}^\top) = \mathrm{CM}(\boldsymbol{z}^\top) \cdot \boldsymbol{1}_N^\top,$$

*for any $\boldsymbol{z} \in \mathbb{L}_\delta$, where $\boldsymbol{\mathcal{L}}_1$ and $\boldsymbol{\mathcal{L}}_2$ denote the token-wise applications of $\mathcal{L}_1$ and $\mathcal{L}_2$, respectively, and $\mathrm{CM}$ is a contextual mapping, which satisfies the following properties:*

1. *For any $\boldsymbol{z} \in \mathbb{L}_\delta$, the $N$ entries in $\mathrm{CM}(\boldsymbol{z}^\top)$ are all distinct.*

2. *For any $\boldsymbol{z}, \boldsymbol{z}' \in \mathbb{L}_\delta$, if $\boldsymbol{z}$ is not a permutation of $\boldsymbol{z}'$, all entries of $\mathrm{CM}(\boldsymbol{z}^\top)$ and $\mathrm{CM}(\boldsymbol{z}'^\top)$ are distinct.*

*Proof.* Due to permutation equivalence, we can assume without loss of generality that elements of $\boldsymbol{z} \in \mathbb{L}_\delta$ is ordered, *i.e.*, $\boldsymbol{z}_1 > \boldsymbol{z}_2 > \cdots > \boldsymbol{z}_N$. Define $\boldsymbol{u} \in \mathbb{R}^N$ as $\boldsymbol{u} := (\delta^{-(N-1)d}, \dots, \delta^{-d}, 1)^\top$, which satisfy

$$\left\| \boldsymbol{u}^\top \boldsymbol{z} - \boldsymbol{u}^\top \boldsymbol{z}' \right\| > 1, \quad \text{if } \boldsymbol{z} \neq \boldsymbol{z}' \text{ for any } \boldsymbol{z}, \boldsymbol{z}' \in \mathbb{L}_\delta.$$

Then $\boldsymbol{z} \to \boldsymbol{u}^\top \boldsymbol{z}$ is a contextual mapping. We show how to construct Transformer block TF : $\mathbb{R}^{3 \times N} \to \mathbb{R}^{3 \times N}$ with single head and head size $s = 1$ such that, for any $\boldsymbol{z} \in \mathbb{L}_\delta$,

$$
\mathrm{TF}^{\circ N}\left(\left[\begin{array}{c} \boldsymbol{z}^\top \\ \boldsymbol{0}_N^\top \\ \boldsymbol{0}_N^\top \end{array}\right]\right) = \left[\begin{array}{c} \boldsymbol{0}_N^\top \\ \boldsymbol{0}_N^\top \\ \boldsymbol{u}^\top \boldsymbol{z} \cdot \boldsymbol{1}_N^\top \end{array}\right].
$$

where $\boldsymbol{0}_N \in \mathbb{R}^N$ denote a zero vector. For $\boldsymbol{z} \in \mathbb{L}_\delta$, we define two series $\boldsymbol{z}^{(k)}$ and $\boldsymbol{s}^{(k)}$ by

$$
\boldsymbol{z}_i^{(k)} = z \quad s.t. \quad \begin{cases} z < 0 & \text{if} \quad i \leq k, \\ z = \boldsymbol{z}_i & \text{otherwise}, \end{cases} \quad \text{for } i = 1, \ldots, d.
$$

$$
\boldsymbol{s}^{(k)} = \sum_{i=0}^{k} \delta^{-id} \boldsymbol{z}_i.
$$

While $\boldsymbol{z}^{(k)}$ is not uniquely determined, any vectors that satisfies the conditions is accepted as $\boldsymbol{z}^{(k)}$. We can confirm that $\boldsymbol{s}^{(k)}$ satisfies

$$
\begin{aligned}
\boldsymbol{s}^{(k)} &= \sum_{i=1}^{k} \delta^{-(k-i)d} \boldsymbol{z}_i \\
&= \Big(\sum_{i=1}^{k-1} \delta^{-(k-i)d} \boldsymbol{z}_i\Big) + \boldsymbol{z}_k \\
&= \Big(\sum_{i=0}^{k-1} \delta^{-d} \cdot \delta^{-(k-1-i)d} \boldsymbol{z}_i\Big) + \boldsymbol{z}_k \\
&= \delta^{-d} \cdot \boldsymbol{s}^{(k-1)} + \boldsymbol{z}_k.
\end{aligned}
\tag{18}
$$

Recall that $\boldsymbol{s}^{(N)} = \boldsymbol{u}^\top \boldsymbol{z}$. Define a single-head self-attention $\mathrm{Attn} : \mathbb{R}^{3 \times N} \to \mathbb{R}^{3 \times N}$ such that

$$
\mathrm{Attn}\left(\left[\begin{array}{c} \boldsymbol{z}^\top \\ \boldsymbol{y}^\top \\ \boldsymbol{s}^\top \end{array}\right]\right) = \left[\begin{array}{c} 0 \\ \max_j \boldsymbol{z}_j \cdot \boldsymbol{1}_N^\top \\ 0 \end{array}\right],
$$

via the weight parameters

$$
\boldsymbol{W}_O = \left[\begin{array}{c} 0 \\ 1 \\ 0 \end{array}\right], \quad \boldsymbol{W}_V = \boldsymbol{W}_K = \boldsymbol{W}_Q = \left[\begin{array}{ccc} 1 & 0 & 0 \end{array}\right]
$$

Define FF : $\mathbb{R}^3 \to \mathbb{R}^3$ of width size $q = 3$ via:

$$
\begin{aligned}
\mathrm{FF}\left(\left[\begin{array}{c} x_1 \\ x_2 \\ x_3 \end{array}\right]\right) &= \left[\begin{array}{ccc} -M & 0 & 0 \\ 0 & -1 & 0 \\ 0 & 1 & \delta^{-d} - 1 \end{array}\right] \sigma_R\left(\left[\begin{array}{ccc} 1 & -1 & 0 \\ 0 & 1 & 0 \\ 0 & 0 & 1 \end{array}\right]\left[\begin{array}{c} x_1 \\ x_2 \\ x_3 \end{array}\right] + \left[\begin{array}{c} \epsilon \\ 0 \\ 0 \end{array}\right]\right) \\
&= \left[\begin{array}{c} -M\sigma_R(x_1 - x_2 + \epsilon) \\ -\sigma_R(x_2) \\ (\delta^{-1} - 1)\sigma_R(x_3) + \sigma_R(x_2) \end{array}\right],
\end{aligned}
$$

where $0 < \epsilon < \delta^{-1}$ and $M > \epsilon \delta^{-1}$. For $x_1 \in \{0, 1, \ldots, \delta^{-d}\}$, we have

$$
x_1 - M\sigma_R(x_1 - x_2 + \epsilon) = z, \quad s.t. \quad \begin{cases} z = x_1 & \text{if} \quad x_1 < x_2, \\ z < 0 & \text{otherwise}. \end{cases}
$$

This post-processes the token ID as if selected, then replaces it with negative value. We have

$$
\begin{aligned}
\boldsymbol{z}_i^{(k)} - M\sigma_R(\boldsymbol{z}_i^{(k)} - \boldsymbol{z}_k + \epsilon) &= \begin{cases} z < 0 & \text{if} \quad i \leq k+1, \\ z = \boldsymbol{z}_i & \text{otherwise}, \end{cases} \\
&= \boldsymbol{z}_i^{(k+1)} \quad \text{for } i = 1, \ldots, d.
\end{aligned}
\tag{19}
$$

We can confirm that Transformer block $\mathrm{TF} : \mathbb{R}^{3 \times N} \to \mathbb{R}^{3 \times N}$ composed of $\mathrm{Attn}$ and $\mathrm{FF}$ satisfies

$$
\mathrm{TF}\left(\begin{bmatrix} (\boldsymbol{z}^{(k)})^\top \\ \mathbf{0}_n^\top \\ (\boldsymbol{s}^{(k)})^\top \end{bmatrix}\right) = (\mathrm{id} + \mathbf{FF}) \circ (\mathrm{id} + \mathrm{Attn})\left(\begin{bmatrix} (\boldsymbol{z}^{(k)})^\top \\ \mathbf{0}_n^\top \\ (\boldsymbol{s}^{(k)})^\top \end{bmatrix}\right)
$$

$$
= (\mathrm{id} + \mathbf{FF})\left(\begin{bmatrix} (\boldsymbol{z}^{(k)})^\top \\ \boldsymbol{z}_k \cdot \mathbf{1}_N^\top \\ (\boldsymbol{s}^{(k)})^\top \end{bmatrix}\right)
$$

$$
= (\mathrm{id} + \mathbf{FF})\left(\begin{bmatrix} (\boldsymbol{z}^{(k)})^\top \\ \boldsymbol{z}_k \cdot \mathbf{1}_N^\top \\ (\boldsymbol{s}^{(k)})^\top \end{bmatrix}\right)
$$

$$
= \begin{bmatrix} (\boldsymbol{z}^{(k)})^\top \\ \boldsymbol{z}_k \cdot \mathbf{1}_N^\top \\ (\boldsymbol{s}^{(k)})^\top \end{bmatrix} + \begin{bmatrix} -M\sigma_R\left((\boldsymbol{z}^{(k)})^\top - \boldsymbol{z}_k \cdot \mathbf{1}_N^\top + \epsilon\mathbf{1}_N^\top\right) \\ -\boldsymbol{z}_k \cdot \mathbf{1}_N^\top \\ (\delta^{-1} - 1)(\boldsymbol{s}^{(k)})^\top + \sigma_R(x_2) \end{bmatrix}
$$

$$
= \begin{bmatrix} (\boldsymbol{z}^{(k+1)})^\top \\ \mathbf{0}_n^\top \\ \delta^{-1}(\boldsymbol{z}^{(k)})^\top + \boldsymbol{z}_k\mathbf{1}_N^\top \end{bmatrix} \quad \text{because Eq. 19}
$$

$$
= \begin{bmatrix} (\boldsymbol{z}^{(k+1)})^\top \\ \mathbf{0}_n^\top \\ (\boldsymbol{s}^{(k+1)})^\top \end{bmatrix} \quad \text{because Eq. 18.}
$$

We define two affine linear maps $\mathcal{L}_1 : \mathbb{R} \to \mathbb{R}^3$ and $\mathcal{L}_2 : \mathbb{R}^3 \to \mathbb{R}$ as $\mathcal{L}_1(x) := (x, 0, 0)$ and $\mathcal{L}_2(x_1, x_2, x_3) := x_3$. Thus, we have

$$
\boldsymbol{\mathcal{L}}_2 \circ \mathrm{TF}^{\circ N} \circ \boldsymbol{\mathcal{L}}_1(x^\top) = \boldsymbol{\mathcal{L}}_2 \circ \mathrm{TF}^{\circ N}\left(\begin{bmatrix} \boldsymbol{z}^\top \\ \mathbf{0}_N^\top \\ \mathbf{0}_N^\top \end{bmatrix}\right) = \boldsymbol{\mathcal{L}}_2\left(\begin{bmatrix} \mathbf{0}_N^\top \\ \mathbf{0}_N^\top \\ \boldsymbol{s}^N \cdot \mathbf{1}_N^\top \end{bmatrix}\right) = (\boldsymbol{u}^\top \boldsymbol{z}) \cdot \mathbf{1}_N^\top.
$$

Recall that $\boldsymbol{z} \to \boldsymbol{u}^\top \boldsymbol{z}$ is a contextual mapping. $\qquad \square$

**Corollary A.6.** *There exist a Transformer block* $\mathrm{TF}_2 : \mathbb{R}^{5 \times N} \to \mathbb{R}^{5 \times N}$ *with single head, head size* $s = 1$*, and width size* $q = 4$*, and two affine linear maps* $\mathcal{L}_1 : \mathbb{R}^2 \to \mathbb{R}^5$ *and* $\mathcal{L}_2 : \mathbb{R}^5 \to \mathbb{R}$ *such that*

$$
\boldsymbol{\mathcal{L}}_2 \circ \mathrm{TF}_2^{\circ N} \circ \boldsymbol{\mathcal{L}}_1\left(\begin{bmatrix} \boldsymbol{z}^\top \\ \boldsymbol{Z}_{d,:} \end{bmatrix}\right) = \left(2\boldsymbol{u}^\top \boldsymbol{z} - \boldsymbol{Z}_{d,N}\right) \cdot \mathbf{1}_N^\top \quad \text{for any } \boldsymbol{z} \in \mathbb{L}_\delta.
$$

*where* $\boldsymbol{u} := (\delta^{-(N-1)d}, \ldots, \delta^{-d}, 1)^\top$.

*Proof.* Define a single-head self-attention $\mathrm{Attn} : \mathbb{R}^{5 \times N} \to \mathbb{R}^{5 \times N}$ such that

$$
\mathrm{Attn}\left(\begin{bmatrix} \boldsymbol{z}^\top \\ \boldsymbol{Z}_{d,:} \\ \boldsymbol{y}^\top \\ \boldsymbol{s}^\top \\ \boldsymbol{q}^\top \end{bmatrix}\right) = \begin{bmatrix} 0 \\ 0 \\ \max_j \boldsymbol{z}_j \cdot \mathbf{1}_N^\top \\ 0 \\ \boldsymbol{Z}_{d,\arg\max_j \boldsymbol{z}_j} \cdot \mathbf{1}_N^\top \end{bmatrix},
$$

via the weight parameters

$$
\boldsymbol{W}_O = \begin{bmatrix} 0 \\ 0 \\ 1 \\ 0 \\ 1 \end{bmatrix}, \quad \boldsymbol{W}_V = \boldsymbol{W}_K = \boldsymbol{W}_Q = \begin{bmatrix} 1 & 0 & 0 & 0 & 0 \end{bmatrix}
$$

Define $\mathrm{FF} : \mathbb{R}^4 \to \mathbb{R}^4$ of width size $q = 4$ via:

$$
\mathrm{FF}\left(\begin{bmatrix} x_1 \\ x_2 \\ x_3 \\ x_4 \\ x_5 \end{bmatrix}\right) = \begin{bmatrix} -M & 0 & 0 & 0 \\ 0 & -1 & 0 & 0 \\ 0 & 1 & \delta^{-d}-1 & 0 \\ 0 & 1 & 0 & -1 \end{bmatrix} \sigma_R \left( \begin{bmatrix} 1 & 0 & -1 & 0 & 0 \\ 0 & 0 & 1 & 0 & 0 \\ 0 & 0 & 0 & 1 & 0 \\ 0 & 0 & 0 & 0 & 1 \end{bmatrix} \begin{bmatrix} x_1 \\ x_2 \\ x_3 \\ x_4 \\ x_5 \end{bmatrix} + \begin{bmatrix} \epsilon \\ 0 \\ 0 \\ 0 \\ 0 \end{bmatrix} \right)
$$

$$
= \begin{bmatrix} -M\sigma_R(x_1 - x_2 + \epsilon) \\ 0 \\ -\sigma_R(x_2) \\ (\delta^{-1} - 1)\sigma_R(x_3) + \sigma_R(x_2) \\ -\sigma_R(x_4) + \sigma_R(x_2) \end{bmatrix},
$$

where $0 < \epsilon < \delta^{-1}$ and $M > \epsilon\delta^{-1}$. Note that the fourth columns after $t$ loops are $z_t$, so at the end, we obtain $z_N$. Then, we define two affine linear maps $\mathcal{L}_1 : \mathbb{R}^2 \to \mathbb{R}^5$ and $\mathcal{L}_2 : \mathbb{R}^5 \to \mathbb{R}$ as $\mathcal{L}_1(x1, x2) := (x1, x2, 0, 0, 0)$ and $\mathcal{L}_2(x_1, x_2, x_3, x_4, x_5) := 2x_4 - x_5$. $\qquad\square$

### A.6 STEP 3. FUNCTION VALUE MAPPING WITH BIT EXTRACTION

We use a bit extraction technique (Bartlett et al., 1998) to approximately memorize the piecewise linear function. Consider $n \in \mathbb{N}$ input indices $k = 0, 1, \ldots, n-1 \in \mathbb{N}$ with associated values $y_0, y_1, \ldots, y_{n-1} \in \mathbb{R}$. The network approximately memorize the difference $y_i - y_{i-1}$ with base-2 representation. Since binary representation limited to $\{0, 1\}$, $y_i - y_{i-1}$ has to be re-scaled with $\epsilon := |y_i - y_{i-1}|$ as

$$
a_i = \lfloor \frac{y_i}{\epsilon} \rfloor,
$$

where $\lfloor x \rfloor = \max\{n : n \le x, \, n \in \mathbb{Z}\}$. Then, the difference $b_i = a_i - a_{i-1}$ satisfies $b_i \in \{-1, 0, 1\}$, $b_i$, and it can be represented using two binary values $c_i, d_i \in \{0, 1\}$ as follows:

$$
b_i = c_i - d_i,
$$

and we have

$$
a_k = a_0 + \sum_{i=0}^{k} b_i = a_0 + \sum_{i=0}^{k} d_i + \sum_{i=0}^{k} d_i \quad \text{for} \quad k = 0, 1, 2.
$$

Lemma A.8 and Lemma 4.1 show that $\sum_{i=0}^{k} c_i$ and $\sum_{i=0}^{k} d_i$ can be realized by composition of single feed-forward layer. Thus the networks can approximate $y_i$ with $\epsilon a_i$ denoted by $\tilde{y}_i$ with the following accuracy

$$
|\tilde{y}_i - y_i| = |\epsilon \lfloor \frac{y_i}{\epsilon} \rfloor - \epsilon \frac{y_i}{\epsilon}| = \epsilon |\lfloor \frac{y_i}{\epsilon} \rfloor - \frac{y_i}{\epsilon}| \le \epsilon.
$$

For $d$-dimensional inputs-outputs pair, we construct the networks for each dimension *i.e.*

$$
\tilde{\boldsymbol{y}} = (\tilde{y}^1, \tilde{y}^2, \ldots, \tilde{y}^d)
$$

The key idea of our lemma and proof follows Lemma D.1 from Zhang et al. (2023) as shown in below; however, we cannot directly apply their result here, as it requires depth-2 networks.

**Proposition A.7** (Lemma D.1 in Zhang et al. (2023)). *Given any $r \in \mathbb{N}^+$, there exist $\mathrm{FF} : \mathbb{R}^{3d} \to \mathbb{R}^{3d}$ of width size 8 and depth 2 with two affine linear maps $\mathcal{L}_1 : \mathbb{R}^2 \to \mathbb{R}^5$ and $\mathcal{L}_2 : \mathbb{R}^5 \to \mathbb{R}$ such that: For any $\theta_1, \theta_2, \ldots, \theta_r \in \{0, 1\}$, it holds that*

$$
\mathcal{L}_2 \circ \mathrm{FF}^{\circ r} \circ \mathcal{L}_1 \big( k, \, \mathrm{bin}\, 0.\theta_1\theta_2 \cdots \theta_r \big) = \sum_{\ell=1}^{k} \theta_\ell \quad \text{for } k = 0, 1, \ldots, r,
$$

*where $\mathrm{bin}\, 0.\theta_1\theta_2 \cdots \theta_r$ denote the binary representation of $\theta = \sum_{i=1}^{n} \theta_i 2^{-i}$.*

We found that loop unrolling allows us to reduce the number of layers from 2 to 1, replacing $x^{k+1} = \mathrm{ReLU}(\mathrm{ReLU}(x'^k))$ with $(x^{k+1}, x'^k) = \mathrm{ReLU}(x'^k, x^k)$. Although our method makes the weights dependent on $\theta_1, \theta_2, \ldots, \theta_r \in 0, 1$, this does not present an issue for our construction in function approximation. Specifically, $\theta_1, \theta_2, \ldots, \theta_r$ is fixed for each target function, and the role of the network is to learn the weights tailored to that single function.

**Lemma A.8.** *Given $\theta_1, \theta_2, \ldots, \theta_r \in \{0, 1\}$ for $r \in \mathbb{N}^+$, there exist a feed-forward layer $\mathrm{FF} : \mathbb{R}^7 \to \mathbb{R}^7$ of width size $10$ and two affine linear maps $\mathcal{L}_1 : \mathbb{R} \to \mathbb{R}^7$ and $\mathcal{L}_2 : \mathbb{R}^7 \to \mathbb{R}$ s.t.*

$$\mathcal{L}_2 \circ (\mathrm{id} + \mathrm{FF})^{\circ r} \circ \mathcal{L}_1(k) = \sum_{i=1}^{k} \theta_i \quad \text{for } k = 0, 1, \ldots, r.$$

*Proof.* From proposition A.2, we have a function $\mathrm{step}(x)$ defined by

$$\mathrm{step}(x) := \sigma_R\left(\tfrac{x}{\epsilon} + 1\right) - \sigma_R\left(\tfrac{x}{\epsilon}\right),$$

satisfies

$$\mathrm{step}(x) = \begin{cases} 1 & \text{if } x \geq 0, \\ 0 & \text{if } x < 0. \end{cases}$$

Define $\beta_i$ for $i = 0, 2, \ldots, r$ as

$$\beta_i = \mathrm{bin}\, 0.\theta_i \cdots \theta_r,$$

where $\mathrm{bin}\, 0.\theta_1\theta_2 \cdots \theta_r$ denote the binary representation of $\theta = \sum_{i=1}^{n} \theta_i 2^{-i}$ and $\theta_0 := 0$. We have

$$\theta_i = \mathrm{step}(\mathrm{bin}\, 0.\theta_i \cdots \theta_r - \tfrac{1}{2}) = \mathrm{step}(\beta_i - \tfrac{1}{2}),$$

implying, for $i = 1, 2, \ldots, r - 1$,

$$\beta_{i+1} = 2\beta_i - \theta_i = 2\beta_i - \mathrm{step}(\beta_i - \tfrac{1}{2}).$$

For given $k \in 0, 1, \ldots, r$, we have

$$\sum_{i=1}^{k} \theta_i = \sum_{i=1}^{k} \theta_i + \sum_{i=k+1}^{r} 0 = \sum_{i=1}^{r} \theta_i \cdot \mathrm{step}(k - i) = \sum_{i=1}^{r} \sigma_R\left(\theta_i + \mathrm{step}(k - i) - 1\right)$$

$$= \sum_{i=1}^{r} \sigma_R\left(\mathrm{step}(\beta_i - \tfrac{1}{2}) + \mathrm{step}(k - i) - 1\right).$$

$$(20)$$

To compute the right-hand side, we need two nested ReLU functions. By using loop unrolling, we precompute $\mathrm{step}(\beta_i - \tfrac{1}{2})$ and $\mathrm{step}(k - i)$ in the previous loops, allowing us to require only a single layer. Define $\boldsymbol{\xi}_l$ for $l = 0, 1, \ldots, r$ as

$$\boldsymbol{\xi}_l = \left(k - l,\ \beta_l,\ \beta_{l+1},\ \mathrm{step}(\beta_l - \tfrac{1}{2}),\ \mathrm{step}(k - l),\ \mathrm{sum}(l)\right)^{\top},$$

where $\mathrm{sum}(l) := \sum_{i=1}^{l} \sigma_R\left(\mathrm{step}(\beta_i - \tfrac{1}{2}) + \mathrm{step}(k - i) - 1\right)$. Note that we have $\beta_{l+1}$ in the $l$-th loop to precompute $\mathrm{step}(\beta_{l+1} - \tfrac{1}{2})$ and $\mathrm{step}\left((k - (l+1))\right)$ for the $l + 1$-th loop.

Define FF : $\mathbb{R}^7 \to \mathbb{R}^7$ such that

$$(\mathrm{id} + \mathrm{FF})(\boldsymbol{\xi}_l) = (\mathrm{id} + \mathrm{FF})\left(\begin{bmatrix} k - l \\ \beta_l \\ \beta_{l+1} \\ \mathrm{step}(\beta_l - \frac{1}{2}) \\ \mathrm{step}(k - l) \\ \mathrm{sum}(l) \end{bmatrix}\right)$$

$$= \begin{bmatrix} k - l \\ \beta_l \\ \beta_{l+1} \\ \mathrm{step}(\beta_l - \frac{1}{2}) \\ \mathrm{step}(k - l) \\ \mathrm{sum}(l) \end{bmatrix} + \begin{bmatrix} 0 & 0 & 0 & 0 & 0 & 0 & 0 & 0 & 0 \\ 1 & 1 & 0 & 0 & 0 & 0 & 0 & 0 & 0 \\ 0 & 0 & 1 & -1 & 1 & 0 & 0 & 0 & 0 \\ 0 & 0 & 0 & 1 & -1 & -1 & 0 & 0 & 0 \\ 0 & 0 & 0 & 0 & -1 & 1 & -1 & 0 & 0 \\ 0 & 0 & 0 & 0 & 0 & 0 & 0 & 0 & 1 \end{bmatrix}$$

$$\sigma_R\left(\begin{bmatrix} 0 & 1 & 0 & 0 & 0 & 0 \\ 0 & 0 & 0 & -1 & 0 & 0 \\ 0 & 0 & 1 & 0 & 0 & 0 \\ 0 & 0 & 1/\epsilon & 0 & 0 & 0 \\ 0 & 0 & 1/\epsilon & 0 & 0 & 0 \\ 0 & 0 & 0 & -1 & 0 & 0 \\ 0 & 0 & 0 & 0 & 1 & 0 \\ 1/\epsilon & 0 & 0 & 0 & 0 & 0 \\ 1/\epsilon & 0 & 0 & 0 & 0 & 0 \\ 0 & 0 & 0 & 1 & 1 & 0 \end{bmatrix}\begin{bmatrix} k - l \\ \beta_l \\ \beta_{l+1} \\ \mathrm{step}(\beta_l - \frac{1}{2}) \\ \mathrm{step}(k - l) \\ \mathrm{sum}(l) \end{bmatrix} + \begin{bmatrix} 0 \\ 0 \\ 0 \\ -1/(2\epsilon) + 1 \\ -1/(2\epsilon) \\ 0 \\ 0 \\ -1/\epsilon + 1 \\ -1/\epsilon \\ -1 \end{bmatrix}\right) + \begin{bmatrix} -1 \\ 0 \\ 0 \\ 0 \\ 0 \\ 0 \\ 0 \end{bmatrix}$$

$$= \begin{bmatrix} k - l \\ \beta_l \\ \beta_{l+1} \\ \mathrm{step}(\beta_l - \frac{1}{2}) \\ \mathrm{step}(k - l) \\ \mathrm{sum}(l) \end{bmatrix} + \begin{bmatrix} -1 \\ \sigma_R(\beta_l) - \sigma_R\big(\mathrm{step}(\beta_l - \frac{1}{2})\big) \\ \sigma_R(\beta_{l+1}) - \big(\sigma_R(\frac{\beta_{l+1} - 1/2}{\epsilon} + 1) - \sigma_R(\frac{\beta_{l+1} - 1/2}{\epsilon})\big) \\ -\sigma_R\big(\mathrm{step}(\beta_l - \frac{1}{2})\big) + \sigma_R(\frac{\beta_{l+1} - 1/2}{\epsilon} + 1) - \sigma_R(\frac{\beta_{l+1} - 1/2}{\epsilon}) \\ -\sigma_R\big(\mathrm{step}(k - l)\big) + \sigma_R(\frac{k - (l+1)}{\epsilon} + 1) - \sigma_R(\frac{k - (l+1)}{\epsilon}) \\ \sigma_R\big(\mathrm{step}(k - l) + \mathrm{step}(\beta_l - \frac{1}{2}) - 1\big) \end{bmatrix}$$

$$= \begin{bmatrix} k - (l + 1) \\ 2\beta_l - \mathrm{step}(\beta_l - \frac{1}{2}) \\ 2\beta_{l+1} - \mathrm{step}(\beta_{l+1} - \frac{1}{2}) \\ \mathrm{step}(\beta_{l+1} - \frac{1}{2}) \\ \mathrm{step}\big((k - (l+1))\big) \\ \mathrm{sum}(l + 1) \end{bmatrix} = \begin{bmatrix} k - (l + 1) \\ \beta_{l+1} \\ \beta_{l+2} \\ \mathrm{step}(\beta_{l+1} - \frac{1}{2}) \\ \mathrm{step}\big((k - (l+1))\big) \\ \mathrm{sum}(l + 1) \end{bmatrix} = \boldsymbol{\xi}_{l+1},$$

Define $\mathcal{L}_1 : \mathbb{R}^2 \to \mathbb{R}^3$ and $\mathcal{L}_2 : \mathbb{R}^3 \to \mathbb{R}$ via

$$\mathcal{L}_1(k) := (k, \beta_0, \beta_1, 0, 0, 0)^\top = \boldsymbol{\xi}_0, \quad \mathcal{L}_2(x_1, x_2, x_3, x_4, x_5, x_6, x_7) := x_7,$$

respectively. Note $\beta_1$ is defined by given $\theta_1, \theta_2, \ldots, \theta_r$ $\qquad \square$

We prove Lemma 4.1 with Lemma A.8.

**Lemma 4.1.** *Given $\boldsymbol{y}_k \in \mathbb{R}^d$ for $k = 0, 1, \ldots, m - 1$ with*

$$|(\boldsymbol{y}_k - \boldsymbol{y}_{k-1})_i| \leq \varepsilon_i \quad \text{for } k = 1, 2, \ldots, m - 1,$$

*there exist feed-forward layer* $\mathrm{FF} : \mathbb{R}^{14d} \to \mathbb{R}^{14d}$ *of width size* $20d$ *and two affine linear maps* $\mathcal{L}_1 : \mathbb{R}^d \to \mathbb{R}^{14d}$ *and* $\mathcal{L}_2 : \mathbb{R}^{14d} \to \mathbb{R}^d$ *such that*

$$\left|\big(\mathcal{L}_2 \circ (\mathrm{id} + \mathrm{FF})^{(m-1)} \circ \mathcal{L}_1(k) - \boldsymbol{y}_k\big)_i\right| \leq \varepsilon_i \quad \text{for } k = 0, 1, \ldots, m - 1,$$

*for any $i = 1, 2, \ldots, d$.*

*Proof.* We prove this for the case where $d = 1$, considering $y_k \in \mathbb{R}$ for $k = 0, \ldots, m$. Define

$$a_i = \left\lfloor \frac{y_i}{\varepsilon} \right\rfloor \quad \text{for } i = 0, 1, \ldots, m - 1,$$

where $\lfloor x \rfloor = \max\{n : n \le x, \ n \in \mathbb{Z}\}$ and set

$$b_i = a_i - a_{i-1} \quad \text{for } i = 1, 2, \ldots, m-1.$$

Since $b_i \in \{-1, 0, 1\}$, there exist $c_i \in \{0, 1\}$ and $d_i \in \{0, 1\}$ such that

$$b_i = c_i - d_i \quad \text{for } i = 1, 2, \ldots, m-1.$$

Thus, we have

$$a_k = a_0 + \sum_{i=1}^{k} c_i - \sum_{i=1}^{k} d_i \quad \text{for } k = 0, 1, \ldots, m-1.$$

From Lemma A.8, there exist $\mathrm{FF}^{(c)}, \mathrm{FF}^{(d)} : \mathbb{R}^7 \to \mathbb{R}^7$ and affine linear maps $\mathcal{L}'_2 : \mathbb{R}^7 \to \mathbb{R}$ and $\mathcal{L}_1^{(c)}, \mathcal{L}_1^{(d)} : \mathbb{R} \to \mathbb{R}^7$ such that

$$\mathcal{L}'_2 \circ \left(\mathrm{id} + \mathrm{FF}^{(c)}\right)^{\circ(m-1)} \circ \mathcal{L}_1^{(c)}(k) = \sum_{i=1}^{k} c_i, \quad \mathcal{L}'_2 \circ \left(\mathrm{id} + \mathrm{FF}^{(d)}\right)^{\circ(m-1)} \circ \mathcal{L}_1^{(d)}(k) = \sum_{i=1}^{k} d_i,$$

for $k = 0, 1, \ldots, m-1$. Then, define $\mathrm{FF} : \mathbb{R}^{14} \to \mathbb{R}^{14}$, for $\boldsymbol{x}, \boldsymbol{y} \in \mathbb{R}^7$,

$$\mathrm{FF}(\boldsymbol{x}, \boldsymbol{y}) := \left(\mathrm{FF}^{(c)}(\boldsymbol{x}), \mathrm{FF}^{(d)}(\boldsymbol{y})\right).$$

Define $\mathcal{L}_1 : \mathbb{R} \to \mathbb{R}^{14}$ and $\mathcal{L}_2 : \mathbb{R}^{14} \to \mathbb{R}$ as

$$\mathcal{L}_1(x) := \left(\mathcal{L}_1^{(c)}(x), \mathcal{L}_1^{(d)}(x)\right), \quad \mathcal{L}_2(\boldsymbol{x}, \boldsymbol{y})^{\top} := \epsilon\left(a_0 + \mathcal{L}'_2(\boldsymbol{x}) - \mathcal{L}'_2(\boldsymbol{y})\right).$$

We can confirm that

$$\mathcal{L}_2 \circ (\mathrm{id} + \mathrm{FF})^{\circ(m-1)} \circ \mathcal{L}_1(k)$$

$$= \mathcal{L}_2 \circ (\mathrm{id} + \mathrm{FF})^{\circ(m-1)} \circ \left(\mathcal{L}_1^{(c)}(k), \ \mathcal{L}_1^{(d)}(k)\right)$$

$$= \mathcal{L}_2 \circ \left((\mathrm{id} + \mathrm{FF}^{(c)})^{\circ(m-1)} \circ \mathcal{L}_1^{(c)}(k), \ (\mathrm{id} + \mathrm{FF}^{(d)})^{\circ(m-1)} \circ \mathcal{L}_1^{(d)}(k)\right)$$

$$= \epsilon\left(a_0 + \mathcal{L}'_2 \circ (\mathrm{id} + \mathrm{FF}^{(c)})^{\circ(m-1)} \circ \mathcal{L}_1^{(c)}(k) - \mathcal{L}'_2 \circ (\mathrm{id} + \mathrm{FF}^{(d)})^{\circ(m-1)} \circ \mathcal{L}_1^{(d)}(k)\right)$$

$$= \epsilon\left(a_0 + \sum_{i=1}^{k} c_i - \sum_{i=1}^{k} d_i\right) = \epsilon a_k.$$

Thus we have

$$\left|\left(\mathcal{L}_2 \circ (\mathrm{id} + \mathrm{FF})^{\circ(m-1)} \circ \mathcal{L}_1(k) - \boldsymbol{y}_k\right)_i\right| = |\epsilon a_k - y_k| \le \varepsilon.$$

We can extend this for $d$-dimensional input. $\qquad\square$

## B ROLE OF TIME DEPENDENT SCALING PARAMETERS

We demonstrate that time-dependent scaling parameters overcome the limitations inherent to the looped architecture and eliminate the dependence of the approximation rate on the modulus of continuity. We use the architecture defined in Section 4 as:

$$\text{FF}(\boldsymbol{x}) \rightarrow \boldsymbol{\eta}(t) \odot \text{FF}(\boldsymbol{x}) \quad \text{for the } t\text{-th loops,}$$

Following lemma demonstrate that time dependent scaling parameters can exactly map indices to output vectors.

**Theorem 4.2.** *Given $\boldsymbol{y}_k \in \mathbb{R}^d$ for $k = 0, \ldots, m-1$, there exist feed-forward layer* $\text{FF} : \mathbb{R}^{4d} \rightarrow \mathbb{R}^{4d}$ *of width size $6d$ and $\boldsymbol{\eta}(t) \in \mathbb{R}^{4d}$ and two affine linear maps $\mathcal{L}_1 : \mathbb{R}^d \rightarrow \mathbb{R}^{4d}$ and $\mathcal{L}_2 : \mathbb{R}^{4d} \rightarrow \mathbb{R}^d$ s.t.*

$$\mathcal{L}_2 \circ (\text{id} + \boldsymbol{\eta}(m-1) \odot \text{FF}) \circ \cdots \circ (\text{id} + \boldsymbol{\eta}(1) \odot \text{FF}) \circ \mathcal{L}_1(k) = \boldsymbol{y}_k.$$

The key idea of the proof is that we use the impulse function defined as

$$\text{impulse}_0((\boldsymbol{y}_l)_i, k-l) = \begin{cases} (\boldsymbol{y}_l)_i & \text{if } k = l, \\ 0 & \text{otherwise,} \end{cases}$$

for $i = 1, 2, \ldots, d$ and $l = 0, 2, \ldots, m-1$, which extracts the corresponding $(\boldsymbol{y}_l)_i$ in the $l$-th loop if the index matches $k$.

*Proof.* We consider the case when $d = 1$, where $y_k \in \mathbb{R}$ for $k = 0, \ldots, m-1$. We update $y_k$ as follows:

$$y_k \rightarrow y_k + \epsilon,$$

where $\epsilon$ is chosen such that none of the $y_l$ values are zero.

Next, we define $\eta(l)$ as:

$$\eta(l) = \left(0, 1, \frac{y_l}{y_{l-1}} - 1, 1\right)^\top \quad \text{for } l = 1, 2, \ldots, m-1.$$

By Proposition A.3, we have

$$\begin{aligned}
\text{impulse}_0(x, t) &:= \sigma_R\big(x + 2M(t + 1/2)\big) - 2M\sigma_R(t + 1/2) \\
&\quad - \sigma_R\big(x + 2M(t - 1/2)\big) + 2M\sigma_R(t - 1/2) \\
&= \begin{cases} x & \text{if } t = 0, \\ 0 & \text{otherwise,} \end{cases}
\end{aligned}$$

where $M > \max y_k$. Define

$$s(l) := \sum_{i=0}^{l} \text{impulse}_0\big(y_{(i-1)}, k - (i-1)\big),$$

for $l = 1, 2, \ldots, m-1$. This satisfies

$$s(m-1) = \sum_{i=0}^{m-1} \text{impulse}_0\big(y_{(i-1)}, k - (i-1)\big) = y_k.$$

Define $\boldsymbol{\xi}_l$ as

$$\boldsymbol{\xi}_l = \Big(k, \ k-l, \ y_l, \ s(l)\Big)^\top.$$

for $l = 0, 1, 2, \ldots, m-1$.

Then, define $\mathrm{FF} : \mathbb{R}^4 \to \mathbb{R}^4$ of width size $q = 6$ via:

$$(\mathrm{id} + \eta(l) \odot \mathrm{FF})(\boldsymbol{\xi}_{l-1}) = \boldsymbol{\xi}_{l-1} +$$

$$\eta(l) \odot \left( \begin{bmatrix} 0 & 0 & 0 & 0 & 0 & 0 \\ 0 & 0 & 0 & 0 & 0 & 0 \\ 1 & -1 & 0 & 0 & 0 & 0 \\ 0 & 0 & 1 & -1 & -2M & 2M \end{bmatrix} \sigma_R \left( \begin{bmatrix} 0 & 0 & 1 & 0 \\ 0 & 0 & -1 & 0 \\ 0 & 2M & 1 & 0 \\ 0 & 2M & 1 & 0 \\ 0 & 1 & 0 & 0 \\ 0 & 1 & 0 & 0 \end{bmatrix} \boldsymbol{\xi}_{l-1} + \begin{bmatrix} 0 \\ -1 \\ M \\ -M \\ 1/2 \\ -1/2 \end{bmatrix} \right) + \begin{bmatrix} 0 \\ -1 \\ 0 \\ 0 \end{bmatrix} \right)$$

$$= \boldsymbol{\xi}_{l-1} + \begin{bmatrix} 0 \\ 1 \\ \frac{y_l}{y_{l-1}} - 1 \\ 1 \end{bmatrix} \odot \begin{bmatrix} 0 \\ -1 \\ \sigma_R(y_{l-1}) - \sigma_R(-y_{l-1}) \\ \left( \sigma_R\big(y_{l-1} + 2M((k - (l-1)) + 1/2)\big) \right. \\ -2M\sigma_R((k - (l-1)) + 1/2) \\ \left. -\sigma_R\big(y_{l-1} + 2M(k - (l-1) - 1/2)\big) + 2M\sigma_R(k - (l-1) - 1/2) \right) \end{bmatrix}$$

$$= \begin{bmatrix} k \\ k - (l-1) \\ y_{l-1} \\ s(l-1) \end{bmatrix} + \begin{bmatrix} 0 \\ 1 \\ \frac{y_l}{y_{l-1}} - 1 \\ 1 \end{bmatrix} \odot \begin{bmatrix} 0 \\ -1 \\ y_{l-1} \\ \mathrm{impulse}_0\big(y_{(l-1)}, k - (l-1)\big) \end{bmatrix}$$

$$= \begin{bmatrix} k \\ k - l \\ y_l \\ s(l) \end{bmatrix} = \boldsymbol{\xi}_l.$$

for $l = 1, 2, \ldots, m - 1$. Thus we have

$$(\mathrm{id} + \eta(m-1) \cdot \mathrm{FF}) \circ \cdots \circ (\mathrm{id} + \eta(1) \cdot \mathrm{FF})(\boldsymbol{\xi}_0) = \boldsymbol{\xi}_{m-1}$$

Then, define two affine linear maps $\mathcal{L}_1 : \mathbb{R}^1 \to \mathbb{R}^4$ and $\mathcal{L}_2 : \mathbb{R}^4 \to \mathbb{R}^1$ by

$$\mathcal{L}_1(x) := (k, k, y_0, 0), \quad \mathcal{L}_2(x_1, x_2, x_3, x_4) := x_4 - \epsilon.$$

We can extend this to $d$-dimensional input by using $d$ time parameters, by applying above to each dimension. $\qquad\square$

## C    DETAILS OF EXPERIMENTS

This appendix section provides additional details on the experiments for each task, including problem descriptions, training configuration, and supplement of results.

### C.1    DYNAMIC PROGRAMMING PROBLEMS

We categorize certain DP problems and employ the following tasks.

- Knapsack Problems: Subset Sum and Knapsack
- Two-Sequence Problems: Edit Distance (ED) and Longest Common Subsequence (LCS)

#### C.1.1    KNAPSACK PROBLEM

We use the knapsack problem and its special case, the subset sum problem. These tasks are solvable in time $\mathcal{O}(nW)$, where $n$ is the length of the input sequence and $W$ represents the weight capacity.

**Subset sum** task is to determine whether there exists a subset of these integers whose sum equals a specified number $T$. It is a subset of the Knapsack problem We randomly sample $n = 10$ integers from the range 1 to 100 and select $T$ randomly from 1 to the sum of these $n$ integers. For example, a sample in the dataset of input sequence length 10 looks like

```
67 93 81 29 2 19 77 74 50 98 | 195 <sep> 1.
```

**Knapsack problem** is defined as follows: given a set of items, each with a weight and a value, select a subset of items that maximizes the total value while ensuring that the total weight does not exceed a specified limit. We concatenated the values, weights, and maximum capacity with a separator. For example, a sample in the dataset of input sequence length 20 looks like

```
9 10 13 1 17 5 12 3 12 2 4 8 11 8 2 10 8 17 10 16 | 8 3 8 7 9 6 7
             2 8 2 3 5 4 2 5 7 10 8 6 7 | 48 <sep> 52.
```

#### C.1.2    TWO SEQUENCES

We use tasks that compute metrics between two given sequences. These tasks can be solved using dynamic programming with a time $\mathcal{O}(n^2)$, where $n$ denotes the length of each input sequence. In the dataset, the two sequences are concatenated with a separator.

**Longest Common Subsequence (LCS)** is the longest common to a given set of sequences. We use problems with input lengths of 60 and 100. Two sequences are sampled uniformly from the alphabet. For example, a sample in the dataset of length 60 looks like

```
g q p b b g q b p h b v i b q m r w c v c v b v w b v g r v q h g
m b r w c r c h i h c c q p m w r w b p g h p w g p w | i i a p i
i i p r i p x i c r b f p b x p i x c c p f r x y i a p c v b p r
        c r v v i c y p x f a c f p p b i i a r a c <sep> 18.
```

**Edit Distance (ED) problem**, also known as Levenshtein distance, is to find the minimum cost required to change one sequence into the other. We adopted the problem setting and data generation approach from Feng et al. (2023), but applied larger input lengths. The costs for insertion, deletion, and replacement were set to 2, 2, and 3, respectively. They generate instances of the edit distance problem as shown in Algorithm 1. The first string is randomly selected, while the second is generated in two ways: (1) a random string yielding a large edit distance, and (2) a corrupted copy of the first string, resulting in a small edit distance. For example, a sample in the dataset of length 60 looks like

```
k y s i s x x x y s s l o o o k o s k o o s l y x k x s y s y x s
o s l y k o o l s k x x y l y i y o s y o y x i i k s | k l l l o
y l y s l k l x i i o k k y o i x s y o s i k x l l x i y k o l y
        o o y x o l s x x l i y l i o s i i i l l y <sep> 105.
```

---

**Algorithm 1:** ED Data Generation from Feng et al. (2023)

---

**Input** : Length of the First String $n$
**Input** : Alphabet $V = \{a, b...z\}$
**Output:** Sequence $s_1, s_2$
Sample $t$ uniformly from $\{3, 4...10\}$ ;
$T \leftarrow$ Sample $t$ letters from $V$ ;
$s_1 \leftarrow$ Sample $n$ letters uniformly from $T$ ;
Sample $p$ uniformly from $[0, 1]$ ;
**if** $p < 0.4$ **then**
    Sample $l$ uniformly from $\{n - 3, n - 2, ..., n + 2\}$;
    $s_2 \leftarrow$ Sample $l$ letters uniformly from $T$ ;
**else**
    **do**
        $s_2 \leftarrow s_1$ ;
        **for** $i \leftarrow 1$ **to** $n$ **do**
            Sample $p$ uniformly from $\{0, 1...len(s_2) - 1\}$;
            Sample $l$ uniformly from $T$;
            Randomly conduct one of the followings: pop $s_2[p]$, substitute $s_2[p]$ with $l$, insert $l$ into $s_2[p]$;
        **end**
    **while** $len(s_2)$ *not in* $[n - 3, n + 2]$;
**end**

---

### C.2 SUPPLEMENTARY INFORMATION ON DP TRAINING AND RESULTS

This section provides supplementary information on the training process and results.

**Training Configuration for DP** We used Looped Transformers of 4 attention heads and a 256-dimensional. We used the AdamW optimizer (Loshchilov & Hutter, 2018) with $\beta_1 = 0.9, \beta_2 = 0.999$, weight decay $= 0.01$, and linear decay scheduler initial lr $= 10^{-4}$ and end lr $= 0$ with 5 warm up, training for 50 epoch with batch size 64. For time-dependent models, we initialize $\gamma(t)$ as zero vectors and $\alpha(t)$ as one vectors, following Peebles & Xie (2023); Bachlechner et al. (2021).

**Training and Test Accuracy for ED** Figure 5 demonstrates a positive correlation between training and test accuracy, allowing us to assess approximation power through test accuracy.

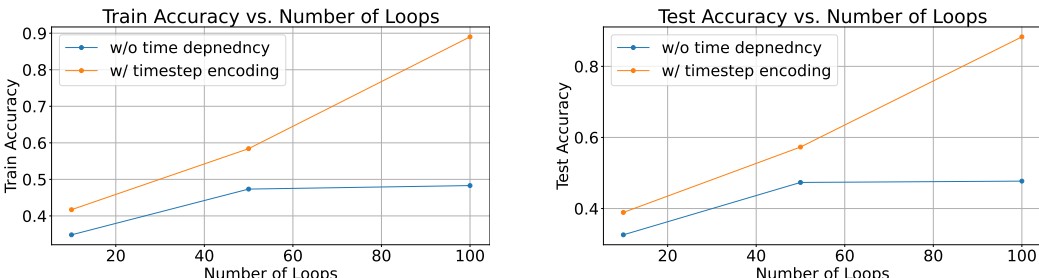

Figure 5: Training and test accuracy for the edit distance task with a sequence length of 60.

### C.3 IN-CONTEXT LEARNING

We followed the setting of Garg et al. (2022); Yang et al. (2024). The problem is to learn the function class from a given sequence composed of the pairs of input $x_i$ and output values $f(x_i)$. The input for model is $(x_1, f(x_1), \ldots, x_k, f(x_k), x_{\text{test}})$, and model learns to predict $f(x_{\text{test}})$. The model is trained on $f(x_k)$ and its performance is evaluated on $f(x_{\text{test}})$ using the squared error.

**Decision Tree.**    We use depth-4 decision trees with 20-dimensional inputs. Each function in this class is represented by a full binary tree with 16 leaf nodes. Non-leaf nodes are associated with specific input coordinates, while leaf nodes are assigned target values. To evaluate $f(x)$, the tree is traversed from the root, moving to the right if the coordinate value is positive and to the left otherwise. Inputs and leaf node values are sampled from $N(0, I)$, and the coordinates for non-leaf nodes are drawn uniformly at random.

**Training Configuration.**    Our training setup follows the approach of Yang et al. (2024). We use Looped Transformers with 8 attention heads and a dimensionality of 256, considering both 12-loop and 70-loop configurations. For time-dependent models, we initialize $\gamma(t)$ as zero vectors and $\alpha(t)$ as one vector. Following the curriculum training approach of Garg et al. (2022); Yang et al. (2024), we progressively increase the task dimensionality from 5 to 20 in steps of 1 every 5000 steps, while the sequence length increases from 26 to 101 in increments of 5 over the same interval. Training is conducted over $200,000$ steps with a learning rate of $1 \times 10^{-4}$.

## C.4    LANGUAGE MODELING

Tokenization is performed using byte-pair encoding, following GPT-2 Radford et al. (2019). The Looped Transformer model is based on the GPT-2 decoder architecture (Radford et al., 2019), with 16 attention heads and a dimensionality of 2048. For time-dependent models, we initialize $\gamma(t)$ as zero vectors and $\alpha(t)$ as one vector. We employed the AdamW optimizer (Loshchilov & Hutter, 2018) with parameters $\beta_1 = 0.9$, $\beta_2 = 0.95$, learning rate lr $= 10^{-4}$, and weight decay weight decay $= 0.1$. The training was conducted for $100000$ iterations with a batch size of 64, a block size of 1024, and 20 gradient accumulation steps. The best perplexity was evaluated on both the training and test sets.

