# OpenReview forum: "On Expressive Power of Looped Transformers: Theoretical Analysis and Enhancement via Timestep Encoding"
_ICLR.cc/2025/Conference — Submitted to ICLR 2025_

### Official Review · Reviewer_MxAi · 2024-11-01

**Soundness:** 2
**Presentation:** 3
**Contribution:** 1
**Rating:** 3
**Confidence:** 3

**Summary:**

This paper theoretically investigates the approximation capabilities of Looped Transformers. Through theoretical analysis, it is found that scaling parameters for each loop can enhance the expressive power of Looped Transformers.

**Strengths:**

One of the strengths of this paper is the detailed theoretical analysis, which is clearly articulated. The definitions of the notation are well presented, making the paper relatively easy to read.

**Weaknesses:**

My main concern lies with the novelty of the work. It is well known that Looped Transformers can be viewed as a type of tied-parameters transformer, and the universal approximation theory for transformers has already been extensively studied. Moreover, the theoretical analysis presented does not yield any novel conclusions regarding the training of Looped Transformers. The finding that performance increases with the number of loops does not seem particularly innovative. Therefore, I have not yet identified any significant conclusions from this work.

My second minor concern is that the theoretical research on Looped Transformers may have lost its research value. Looped Transformers were proposed to address tasks that transformers struggle with, which RNNs can accomplish easily, while also offering parameter efficiency. However, transformers have now completely supplanted RNNs in various domains through various training techniques. In the most critical application areas for transformers, such as autoregressive models and LLMs, Looped Transformers are not commonly used, and their performance does not match that of transformer architectures.

**Questions:**

See weakness

---

> ### Author Response · Authors · 2024-11-19
> **Rebuttal by Authors (1/2)**
>
> We sincerely appreciate your time and thoughtful review of our paper. Please find our responses to your questions below.
>
> > Q1: My main concern lies with the novelty of the work. It is well known that Looped Transformers can be viewed as a type of tied-parameters transformer, and the universal approximation theory for transformers has already been extensively studied.
>
> A1: **Our paper is the first to establish the approximation rate of Looped Transformers, introducing three novel types of modulus of continuity newly defined for sequence-to-sequence functions**. Due to the structural constraints imposed by weight-tying, which limit their flexibility, **universal approximation theories for Transformers cannot be directly applied**. Consequently, even proving the universal approximation theorem for fixed single-layer Looped Transformers is non-trivial.
>
> For the case of ReLU networks, Zhang et al. [1] provided significant insights into the expressive power of weight-tied ReLU layers. While the universality of ReLU networks has been extensively studied, their findings regarding the surprising capacity of a single fixed-size ReLU layer and the potential for enhancement through composition were particularly novel.
>
> The approximation rate provides a clear theoretical understanding of how the error decreases as the number of loops increases, its dependency on data, and the rate of reduction. It enables better model design with theoretical guarantees.
>
> [1] Zhang et al., "On Enhancing Expressive Power via Compositions of Single Fixed-Size ReLU Network," ICML 2023.
>
> We added the following explanation to clarify our novelty.
>
> **line 41**: Due to the structural constraints imposed by weight-tying, which limit their flexibility, existing universal approximation theories for Transformers cannot be directly applied.
>
> > Q2: Moreover, the theoretical analysis presented does not yield any novel conclusions regarding the training of Looped Transformers. The finding that performance increases with the number of loops does not seem particularly innovative. Therefore, I have not yet identified any significant conclusions from this work.
>
> A2: Beyond establishing the approximation rate, our work identifies a specific **limitation inherent to the looped architecture**. To address this, we introduce the incorporation of scaling parameters for each loop, implemented with conditioning on timestep encoding (see Sec.4). We demonstrate the **enhancement via timestep encoding** both theoretically and experimentally.
> While the result that performance improves with more loops may seem straightforward, the approximation rate we derive, along with its dependency on the continuity properties of sequence-to-sequence functions, represents a non-trivial contribution.
>
> In addition, to validate the enhancement via timestep encoding, we add empirical evidence on in-context learning (decision tree function class) and language modeling (WikiText103).
>
> ### In-Context Learning (MSE(↓) for Decision Trees)
>
> | TF (d=12) | Looped TF (r=12) | w/ Timestep (r=12) |
> |----------------------|----------------------|----------------------|
> | 8.64e-03    | 1.43e-02                | **1.70e-03**           |
>
> ### Language Modeling (WikiText-103)
>
> | Metric                | TF (d=12) | Looped TF (r=1) | Looped TF (r=3) | Looped TF (r=6) | w/ Timestep Encoding (r=1) | w/ Timestep Encoding (r=3) | w/ Timestep Encoding (r=6) |
> |-----------------------|-----------|-----------------|-----------------|-----------------|----------------------------|----------------------------|----------------------------|
> | Train Perplexity (↓) | 5.11      | 6.65           | 5.64           | 5.61           | 6.29                       | 5.31                       | **5.05**                   |
> | Test Perplexity (↓)  | **19.6**  | 33.11          | 27.93          | 28.16          | 31.18                      | 23.45                      | 22.42                      |

---

> ### Author Response · Authors · 2024-11-19
> **Rebuttal by Authors (2/2)**
>
> > Q3: My second minor concern is that the theoretical research on Looped Transformers may have lost its research value…. In the most critical application areas for transformers, such as autoregressive models and LLMs, Looped Transformers are not commonly used, and their performance does not match that of transformer architectures.
>
> A3: Looped Transformers hold their research value because of the following three points.
>
> 1. [**Extensive Practical Applications and Research Interest.**] Looped Transformers have been practically studied for efficient and competitive capabilities [1, 2] and remain an active area of research. Previous works have studied their Turing completeness [3], generalization ability [4, 5], inductive bias for reasoning tasks [6], ability of learning iterative algorithms for in-context learning [7, 8, 9], and potential to simulate graph algorithms [10].
>
>    To highlight the advantages of Looped Transformers, we included the following sentence in the introduction section. We cover related works in subsection 2.2
>
>    **line 24**: This structure offers advantages over standard Transformers, such as inductive bias, parameter efficiency, and Turing completeness.
>
> 2. [**Distinct from RNNs.**] Looped Transformers differ from RNNs. The recurrent structure is fundamentally distinct. In particular, Looped Transformers retain the parallelism that has been a critical advantage driving the adoption of Transformers over RNNs.
>
> 3. [**Potentially Advantage for Complex Tasks.**] Our experiments demonstrated that Looped Transformers solve certain dynamic programming problems, which standard Transformers struggle with (as shown below). Given the current trend of moving from language tasks to more complex ones like mathematical theorem proving, Looped Transformers may hold increasing value.
>
> | **Task**        | **TF** | **Looped (r=5)** | **Looped (r=10)** | **Looped (r=50)** | **Looped (r=100)** | **Timestep (r=5)** | **Timestep (r=10)** | **Timestep (r=50)** | **Timestep (r=100)** |
> |------------------|--------|------------------|-------------------|-------------------|--------------------|--------------------|---------------------|---------------------|----------------------|
> | Subset Sum (10) | 83.4   | **84.1**         | 83.0             | -                 | -                  | 83.8               | 83.9               | -                   | -                    |
> | Knapsack (20)   | 92.8   | 92.2             | **94.0**         | -                 | -                  | 88.7               | 90.9               | -                   | -                    |
> | LCS (60)        | 70.0   | 66.0             | 81.8             | 98.6             | -                  | 68.5               | 80.5               | **99.3**           | -                    |
> | LCS (100)       | 39.8   | 39.6             | 45.1             | 93.5             | -                  | 36.7               | 45.6               | **98.1**           | -                    |
> | ED (60)         | 41.4   | 23.8             | 32.6             | 47.3             | 47.7              | 26.6               | 38.9               | 57.3               | **88.3**            |
>
> In research, we believe that significance lies not only in models already in use but also in exploring a broad range of models that may potentially become important in the future.
>
> [1] Lan et al., ALBERT: A Lite BERT for Self-supervised Learning of Language Representations. ICLR 2020.
> [2] Bae et al., Relaxed Recursive Transformers: Effective Parameter Sharing with Layer-wise LoRA. 2024.
> [3] Giannou et al., Looped Transformers as Programmable Computers. ICML 2023.
> [4] Dehghani et al., Universal Transformers, ICLR 2019.
> [5] Fang et al., Looped Transformers for Length Generalization. 2024.
> [6] Anonymous, Understanding Reasoning with Looped Models. ICLR 2025 under review.
> [7] Yang et al., Looped Transformers are Better at Learning Learning Algorithms. ICLR 2024.
> [8] Giannou et al., How Well Can Transformers Emulate In-context Newton's Method? 2024.
> [9] Gatmiry et al., Can Looped Transformers Learn to Implement Multi-step Gradient Descent for In-context Learning? ICML 2024.
> [10] Luca and Fountoulakis, Simulation of Graph Algorithms with Looped Transformers. ICML 2024.

---

> ### Author Response · Authors · 2024-11-28
> **Seeking Feedback on Rebuttal Response by Authors**
>
> Dear Reviewer MxAi,
>
> Thank you for taking the time to review our paper and provide valuable feedback. We have submitted a rebuttal addressing the issues and suggestions you kindly pointed out. Could you confirm if our response has adequately resolved the concerns you raised? If there are any areas where our response may still be insufficient, we would greatly appreciate further clarification or guidance, as we are eager to engage in a constructive discussion.
>
> Your insights are essential for improving the quality of our paper, and we sincerely look forward to hearing from you.
>
> Best regards

---

### Official Review · Reviewer_C5sB · 2024-11-03

**Soundness:** 3
**Presentation:** 4
**Contribution:** 3
**Rating:** 8
**Confidence:** 4

**Summary:**

The paper presents a theoretical exploration of the expressive power of Looped Transformers, particularly regarding their function approximation capabilities. The study justifies that Looped Transformer, while offering parameter efficiency and Turing completeness, suffers from some limitations in the approximation rate, and it addresses it by introducing timestep encoding, which enhances expressive capacity by conditioning on time-dependent scaling factors.

**Strengths:**

1.	The paper is well-written. The problem formulation, theoretical analysis, and experimental results are described clearly.
2.	The paper's contribution is relatively significant. It derives bounds on the approximation rate of a looped transformer to demonstrate the limited approximation ability.
3.	The paper provides strong theoretical justifications that lead to its design (of timestep encoding) on loop transformers. In particular, looped architecture is revealed to have exponential asymptotic approximation rate decay, which leads to the incorporation of timestep encoding.

**Weaknesses:**

1.	The experimental validation is relatively limited. In particular, only one task is evaluated on the timestep encoded model compared with the baseline. The authors are encouraged to test on more tasks including basic ones (e.g. inductive heads) for better validation of the model improvement and use ablation studies for more detailed characterization.

**Questions:**

1.	How tight is the approximation rate bound derived from the paper? It would be interesting to see empirical results that validate equation (9) or (10).

---

> ### Author Response · Authors · 2024-11-19
> **Rebuttal by Authors**
>
> We sincerely appreciate your time and thoughtful review of our paper. Please find our responses to your questions below.
>
> > Q1: The experimental validation is relatively limited. In particular, only one task is evaluated on the timestep encoded model compared with the baseline. The authors are encouraged to test on more tasks including basic ones (e.g. inductive heads) for better validation of the model improvement and use ablation studies for more detailed characterization.
>
> A1: Thank you for your helpful suggestions. We expanded 5 experiments include additional dynamic programming problems (subset sum, knapsack, LCS), in-context learning (decision tree function class), analogous to inductive heads, and language modeling using the WikiText-103 dataset. We are conducting additional experiments and will report the results in the discussion thread if they are ready in time.
>
> ### Dynamic Programming (Accuracy (↑))
>
> | **Task**        | **TF** | **Looped (r=5)** | **Looped (r=10)** | **Looped (r=50)** | **Looped (r=100)** | **Timestep (r=5)** | **Timestep (r=10)** | **Timestep (r=50)** | **Timestep (r=100)** |
> |------------------|--------|------------------|-------------------|-------------------|--------------------|--------------------|---------------------|---------------------|----------------------|
> | Subset Sum (10) | 83.4   | **84.1**         | 83.0             | -                 | -                  | 83.8               | 83.9               | -                   | -                    |
> | Knapsack (20)   | 92.8   | 92.2             | **94.0**         | -                 | -                  | 88.7               | 90.9               | -                   | -                    |
> | LCS (60)        | 70.0   | 66.0             | 81.8             | 98.6             | -                  | 68.5               | 80.5               | **99.3**           | -                    |
> | LCS (100)       | 39.8   | 39.6             | 45.1             | 93.5             | -                  | 36.7               | 45.6               | **98.1**           | -                    |
> | ED (60)         | 41.4   | 23.8             | 32.6             | 47.3             | 47.7              | 26.6               | 38.9               | 57.3               | **88.3**            |
>
> ### In-Context Learning (MSE(↓) for Decision Trees)
>
> | TF (d=12) | Looped TF (r=12) | w/ Timestep (r=12) |
> |----------------------|----------------------|----------------------|
> | 8.64e-03    | 1.43e-02                | **1.70e-03**           |
>
> ### Language Modeling (WikiText-103)
>
> | Metric                | TF (d=12) | Looped TF (r=1) | Looped TF (r=3) | Looped TF (r=6) | w/ Timestep Encoding (r=1) | w/ Timestep Encoding (r=3) | w/ Timestep Encoding (r=6) |
> |-----------------------|-----------|-----------------|-----------------|-----------------|----------------------------|----------------------------|----------------------------|
> | Train Perplexity (↓) | 5.11      | 6.65           | 5.64           | 5.61           | 6.29                       | 5.31                       | **5.05**                   |
> | Test Perplexity (↓)  | **19.6**  | 33.11          | 27.93          | 28.16          | 31.18                      | 23.45                      | 22.42                      |
>
> > Q2: How tight is the approximation rate bound derived from the paper? It would be interesting to see empirical results that validate equation (9) or (10).
>
> A2: Thank you for your insightful question. The tightness of the approximation rate remains an open issue. Although we can state that it aligns with the findings of prior studies on ReLU networks [1], designing sequence-to-sequence functions while controlling the three types of continuity is challenging. Furthermore, even the tightness of ReLU has not been thoroughly investigated. This will be addressed as a direction for future research.
>
> [1] Zhang et al., "On Enhancing Expressive Power via Compositions of Single Fixed-Size ReLU Network," ICML 2023.
>
> We add this limitation in Conclusions as follows.
> **line 537:** While we have derived upper bounds, the tightness of these approximation rates is still undetermined.

---

> > ### Comment · Reviewer_C5sB · 2024-11-25
> >
> > Thank you for the response. The rebuttal has addressed all my questions.

---

### Official Review · Reviewer_H7Gd · 2024-11-04

**Soundness:** 3
**Presentation:** 2
**Contribution:** 3
**Rating:** 6
**Confidence:** 4

**Summary:**

This paper delves into the approximation rate of Looped Transformers for continuous sequence-to-sequence functions, offering a comprehensive mathematical analysis of the strengths and limitations of looped architectures. Specifically, it examines Looped Transformers by defining critical concepts such as sequence, contextual, and token continuity, which significantly influence the approximation rate. The authors propose a single-loop topology for contextual mapping that requires only a single-layer network for universal approximation.
This paper's novel contribution is the introduction of a new time-stepping encoding method, which addresses the approximation errors typically associated with Looped Transformers. This method incorporates a time-dependent scaling parameter, effectively discretizing variations in the target function. The study demonstrates that time-dependent Looped Transformer models can mitigate approximation errors arising from contextual and token continuity issues.
Experimental results support these claims, showing that accuracy increases linearly with the number of loops when using timestep-encoded models, in contrast to time-dependent models, which tend to saturate in training and accuracy. Additionally, the proposed approach offers resource optimization by reducing the number of layers in the neural network and decreasing the number of parameters by approximately 90% compared to previous works.

**Strengths:**

The presented methodology introduces a novel approach that effectively addresses the known limitations of Looped Transformer model architectures. The paper provides a robust background, thoroughly reviewing previous studies in the field. It includes rigorous mathematical demonstrations that substantiate the claims, and the experimental results are consistent with these theoretical findings.

**Weaknesses:**

The redaction can be challenging to follow in certain sections, particularly where large equations are interspersed with text (e.g., Theorem 3.7, Lemma 4.2). To enhance clarity, it would be beneficial to include comparison charts against current Looped Transformer models. These charts would help to emphasize the advantages of the proposed methodology and illustrate the behavior observed across different models.

**Questions:**

Are you planning to scale up experiments by using different models? Is it anticipated to be feasible to apply the proposed architecture to different scenarios? Are there any future work plans?

---

> ### Author Response · Authors · 2024-11-19
> **Rebuttal by Authors**
>
> We sincerely appreciate your time and thoughtful review of our paper. Please find our responses to your questions below.
>
> > Q1: To enhance clarity, it would be beneficial to include comparison charts against current Looped Transformer models. These charts would help to emphasize the advantages of the proposed methodology and illustrate the behavior observed across different models.
>
> A1: To enhance clarity, we add the chart of our timestep encoding methodology as Figure 2: Timestep encoding architecture.
> The difference is relatively straightforward, so the pseudocode below provides a simple comparison, which I hope will be helpful for understanding.
>
> ### Standard Block
>
> ```python
> def forward(x):
>     x = x + attn(norm1(x))
>     x = x + mlp(norm2(x))
>     return x
> ```
>
> ### w/ Timestep Encoding
> ```python
> def forward(x, t):
>     t_emb = timstep_enc(t)
>     scale_msa, scale_mlp, gate_msa, gate_mlp = ada_mlp(t_emb).chunk(4)
>     x = x + gate_msa * attn(norm1(x) * (scale_msa)
>     x = x + gate_mlp.unsqueeze(1) * mlp(norm2(x) * scale_mlp)
>     return x
> ```
>
>
> > Q2: Are you planning to scale up experiments by using different models? Is it anticipated to be feasible to apply the proposed architecture to different scenarios? Are there any future work plans?
>
> A2: Thanks to your suggestion, we expanded 5 tasks: dynamic programming problems (subset sum, knapsack, LCS), in-context learning (decision tree function class), and language modeling using the Wikitext-103 dataset.
> We are conducting additional experiments and will report the results in the discussion thread if they are ready in time.
>
> ### Dynamic Programming (Accuracy (↑))
>
> | **Task**        | **TF** | **Looped (r=5)** | **Looped (r=10)** | **Looped (r=50)** | **Looped (r=100)** | **Timestep (r=5)** | **Timestep (r=10)** | **Timestep (r=50)** | **Timestep (r=100)** |
> |------------------|--------|------------------|-------------------|-------------------|--------------------|--------------------|---------------------|---------------------|----------------------|
> | Subset Sum (10) | 83.4   | **84.1**         | 83.0             | -                 | -                  | 83.8               | 83.9               | -                   | -                    |
> | Knapsack (20)   | 92.8   | 92.2             | **94.0**         | -                 | -                  | 88.7               | 90.9               | -                   | -                    |
> | LCS (60)        | 70.0   | 66.0             | 81.8             | 98.6             | -                  | 68.5               | 80.5               | **99.3**           | -                    |
> | LCS (100)       | 39.8   | 39.6             | 45.1             | 93.5             | -                  | 36.7               | 45.6               | **98.1**           | -                    |
> | ED (60)         | 41.4   | 23.8             | 32.6             | 47.3             | 47.7              | 26.6               | 38.9               | 57.3               | **88.3**            |
>
> ### In-Context Learning (MSE(↓) for Decision Trees)
>
> | TF (d=12) | Looped TF (r=12) | w/ Timestep (r=12) |
> |----------------------|----------------------|----------------------|
> | 8.64e-03    | 1.43e-02                | **1.70e-03**           |
>
> ### Language Modeling (WikiText-103)
>
> | Metric                | TF (d=12) | Looped TF (r=1) | Looped TF (r=3) | Looped TF (r=6) | w/ Timestep Encoding (r=1) | w/ Timestep Encoding (r=3) | w/ Timestep Encoding (r=6) |
> |-----------------------|-----------|-----------------|-----------------|-----------------|----------------------------|----------------------------|----------------------------|
> | Train Perplexity (↓) | 5.11      | 6.65           | 5.64           | 5.61           | 6.29                       | 5.31                       | **5.05**                   |
> | Test Perplexity (↓)  | **19.6**  | 33.11          | 27.93          | 28.16          | 31.18                      | 23.45                      | 22.42                      |
>
>
> > Q3: The redaction can be challenging to follow in certain sections, particularly where large equations are interspersed with text (e.g., Theorem 3.7, Lemma 4.2).
>
> A3: The layout is currently constrained by space limitations; however, we will make effort to address this issue and improve readability.

---

> ### Author Response · Authors · 2024-11-28
> **Seeking Feedback on Rebuttal Response by Authors**
>
> Dear Reviewer H7Gd,
>
> Thank you for taking the time to review our paper and provide valuable feedback. We have submitted a rebuttal addressing the issues and suggestions you kindly pointed out. Could you confirm if our response has adequately resolved the concerns you raised? If there are any areas where our response may still be insufficient, we would greatly appreciate further clarification or guidance, as we are eager to engage in a constructive discussion.
>
> Your insights are essential for improving the quality of our paper, and we sincerely look forward to hearing from you.
>
> Best regards

---

### Author Response · Authors · 2024-12-02
**Summary of Rebuttal by Authors (1/2)**

Dear Area Chairs,

We have carefully addressed and respectfully refuted the questions raised by the reviewers, as summarized below. Although we have yet to receive feedback from some reviewers, we believe that all concerns have been thoughtfully considered and appropriately resolved to the best of our ability.

***

> Q1: (Reviewer MxAi) My main concern lies with the novelty of the work. It is well known that Looped Transformers can be viewed as a type of tied-parameters transformer, and the universal approximation theory for transformers has already been extensively studied. Moreover, the theoretical analysis presented does not yield any novel conclusions regarding the training of Looped Transformers. The finding that performance increases with the number of loops does not seem particularly innovative. Therefore, I have not yet identified any significant conclusions from this work.

A1: **Our paper is the first to establish the approximation rate of Looped Transformers, introducing three types of modulus of continuity newly defined for sequence-to-sequence functions**. Due to the structural constraints imposed by weight-tying, which limit their flexibility, **universal approximation theories for Transformers cannot be directly applied**.

***

> Q2: (Reviewer MxAi) Moreover, the theoretical analysis presented does not yield any novel conclusions regarding the training of Looped Transformers. The finding that performance increases with the number of loops does not seem particularly innovative. Therefore, I have not yet identified any significant conclusions from this work.

A2: **Beyond establishing the approximation rate, our work identifies a specific limitation inherent to the looped architecture**. To address this, we introduce the incorporation of scaling parameters for each loop, implemented with conditioning on timestep encoding (see Sec.4). We demonstrate the **enhancement via timestep encoding** both theoretically and experimentally.

***

> Q3: (Reviewer MxAi) My second minor concern is that the theoretical research on Looped Transformers may have lost its research value…. In the most critical application areas for transformers, such as autoregressive models and LLMs, Looped Transformers are not commonly used, and their performance does not match that of transformer architectures.

A3: Looped Transformers hold their research value because of the following three points.

(1). **Extensive Practical Applications and Research Interest**: Looped Transformers have been practically studied for efficient and competitive capabilities [1, 2] and remain an active area of research. Previous works have studied their Turing completeness [3], generalization ability [4, 5], inductive bias for reasoning tasks [6], ability of learning iterative algorithms for in-context learning [7, 8, 9], and potential to simulate graph algorithms [10]. (All the reference is shown in rebuttal to Reviewer MxAi)

We also emphasized (2) **the distinctiveness from RNNs** and (3). **the potential advantages for complex tasks**.

***

> Q4 (Reviewer H7Gd): The redaction can be challenging to follow in certain sections, ...

A4: To enhance clarity, we add the chart of our timestep encoding methodology as Figure 2.

***

> Q5: (Reviewer H7Gd) Are you planning to scale up experiments by using different models? (Reviewer C5sB) The experimental validation is relatively limited.

A5: **We expanded 5 tasks**: dynamic programming problems (subset sum, knapsack, LCS), in-context learning (decision tree), and language modeling (Wikitext-103).

---

> ### Author Response · Authors · 2024-12-02
> **Summary of Rebuttal by Authors (2/2)**
>
> # Experimental results
>
> ## Dynamic Programming (Accuracy (↑))
>
> | **Task**        | **TF** | **Looped (r=5)** | **Looped (r=10)** | **Looped (r=50)** | **Looped (r=100)** | **Timestep (r=5)** | **Timestep (r=10)** | **Timestep (r=50)** | **Timestep (r=100)** |
> |------------------|--------|------------------|-------------------|-------------------|--------------------|--------------------|---------------------|---------------------|----------------------|
> | Subset Sum (10) | 83.4   | **84.1**         | 83.0             | -                 | -                  | 83.8               | 83.9               | -                   | -                    |
> | Knapsack (20)   | 92.8   | 92.2             | **94.0**         | -                 | -                  | 88.7               | 90.9               | -                   | -                    |
> | LCS (60)        | 70.0   | 66.0             | 81.8             | 98.6             | -                  | 68.5               | 80.5               | **99.3**           | -                    |
> | LCS (100)       | 39.8   | 39.6             | 45.1             | 93.5             | -                  | 36.7               | 45.6               | **98.1**           | -                    |
> | ED (60)         | 41.4   | 23.8             | 32.6             | 47.3             | 47.7              | 26.6               | 38.9               | 57.3               | **88.3**            |
>
> ## In-Context Learning (MSE(↓) for Decision Trees)
>
> | TF (d=12) | Looped TF (r=12) | w/ Timestep (r=12) |
> |----------------------|----------------------|----------------------|
> | 8.64e-03    | 1.43e-02                | **1.70e-03**           |
>
> ## Language Modeling (WikiText-103)
>
> | Metric                | TF (d=12) | Looped TF (r=1) | Looped TF (r=3) | Looped TF (r=6) | w/ Timestep Encoding (r=1) | w/ Timestep Encoding (r=3) | w/ Timestep Encoding (r=6) |
> |-----------------------|-----------|-----------------|-----------------|-----------------|----------------------------|----------------------------|----------------------------|
> | Train Perplexity (↓) | 5.11      | 6.65           | 5.64           | 5.61           | 6.29                       | 5.31                       | **5.05**                   |
> | Test Perplexity (↓)  | **19.6**  | 33.11          | 27.93          | 28.16          | 31.18                      | 23.45                      | 22.42                      |

---

### Meta-Review · Area_Chair_tYah · 2024-12-24

**Metareview:**

**Summary:**
This paper studies expressive ability and approximation rate of the looped transformers.

**Strengths:**
This paper provides a theoretical analysis on expressive abilities of the looped transformers, claiming that it has revealed limitations specific to the looped architecture. On the basis of the theoretical findings, this paper furthermore proposes introduction of timestep encoding with the aim of improving the approximation rate.

**Weaknesses:**
The validity of the arguments in the theoretical part of this paper is questionable. See below.

**Reasons:**
I realized in the very final phase of reviewing that there appear several serious flaws in Theorem 3.7 and its proof, which constitutes the main contribution of this paper. Unfortunately, it seems that these flaws have not been noticed by any reviewers, most likely because they are not asked to read supplementary materials. Given these flaws, I would judge the technical quality of this paper not to reach the level of the conference, and would refrain from recommending acceptance.

The following part details those flaws.

**On $r$:** It is claimed that the statement of Theorem 3.7 holds "for any $r\in\mathbb{N}$" (line 203), but it cannot. (a) If $r$ is less than or equal to $N$, then $\delta=((r-N)/2)^{-(N+1)d-1}$ is not positive, causing problems. (b) If the for-any claim be valid, then one might let $r=1$, implying that the statement holds for non-looped transformers, making the discussion on looped transformers meaningless.

**On Equation (10):** It is claimed that the conditions $\delta^{-1}+2\delta^{-(N+1)d}=r-N$ (line 366) and $\delta^{-1}\cdot2\delta^{-(N+1)d}\ge r-N$ (line 367) on $\{\delta,N,d,r\}$ are equivalent, but this claim is not correct. It makes the development here, as well as the whole statement of Theorem 3.7, invalid.

**On $\epsilon$:** As shown in Propositions A.1 and A.2, compositions of ReLU functions approximate the rectangular and step functions, only in the limit $\epsilon\to0$. In the proof of Lemma A.4, however, this was not treated appropriately. The formula in line 1055, as well as the claim of Lemma A.4, is valid only in the limit $\epsilon\to0$.

One also observes that some weights will diverge in the limit $\epsilon\to0$: see the formula in lines 1087-1092. The unboundedness of the weights may be problematic, especially in contrast to the well-known Barron-type universal approximation theorems, where the universal approximation property holds with parameters whose values are bounded (see, e.g., Caragea, Petersen, Voigtlaender, Ann. Appl. Probab. 33(4): 3039-3079, 2023). In any case, the authors should discuss $\epsilon$-dependence of their approximation rate, as well as dependence of bounds of the weights on $\epsilon$.

In addition to the points mentioned above, there are numerous points which would require revision, although each of them might not be fatal. I would like to encourage the authors to proofread their manuscript much more carefully. These are listed in **Additional comments** section below due to the space constraint.

**Additional Comments On Reviewer Discussion:**

Although two out of the three reviewers evaluated this paper positively, because of the numerous apparent flaws noticed (see above), I would judge the technical quality of this paper not to reach the standard of the conference.

**Detailed comments:**
- Definition 3.3: The range of the function $f$ should be specified as a normed vector space. The left-hand side of the formula should be $\\\|f\\\|_{L^p([0,1]^{d\times N})}$.
- Definition 3.5: The notation $(\cdot)_{:,n}$ should be defined at its first appearance.
- Line 186: can induce(s)
- Lines 261: Put a space just before "however,".
- Line 262: denote(d); ordered $N$ token IDs → $N$ ordered token IDs
- Lines 266, 771, 1157: which satisf(y → ies)
- Line 292: the constrain(t)
- Line 315: they require(s)
- Lines 324, 765: $C\delta^d$ is not the volume itself but its upper bound.
- Line 363: $\delta^{-1}-1$ → $\delta^{-1}$
- Line 401: as (show in → shown) below
- Lines 411-412: Strictly speaking, as opposed to the authors' claim, Lemma 4.1 does not imply that large values of $|(y_k-y_{k-1})_i|$ would result in increase in the approximation error (i.e., $A\to B$ does not imply $\neg A\to\neg B$). Thus, one can question the claimed cause of dependence.
- Lines 405, 426, 780, 794, 903, 1026, 1238: there exist(s)
- Line 696: $ℬ_{i,n}\delta+1$ → $(ℬ_{i,n}+1)\delta$
- Line 701: What value of $t$ one should choose have to be specified. $1_{d\times d}$ → $1_{d\times N}$
- Line 705: $Q$ of $Q_{\mathcal{B}}$ should be in boldface.
- Line 712: Although this inequality is valid, what is used in equation (14) is not this finite-dimensional version, but a functional-space version.
- Line 721: (the) three step(s)
- Line 724: (divided → discretized)
- Line 734: (are → is) divided into
- Lines 736, 741: $\beta_i\delta+1$ → $(\beta_i+1)\delta$
- Lines 741, 754, 852, 913, 1030, 1113: $\circ(\delta^{-1}-1)$ → $\circ\delta^{-1}$
- Line 749: The range of $\mathcal{L}_2^{(1)}$ is not $\mathbb{R}^d$ but $\mathbb{R}$. The range of $\mathcal{L}_2'^{(1)}$ is not $\mathbb{R}$ but $\mathbb{R}^d$.
- Lines 767, 828: distinct $N$ → $N$ distinct
- Lines 770, 1128, 1155: permutation equiva(le → ria)nce
- Line 792: The redundant pair of parentheses should be removed.
- Line 794: $\mathrm{TF}_2'$ should perhaps read $\mathrm{TF}'^{(2)}$.
- Line 826: The base used in the formula in lines 806-809 would be not $\delta^{-d}$ but $2\delta^{-d}$.
- Line 830: $\mathcal{L}_3$ and $\mathcal{K}_1$ are undefined.
- Line 839: contain(s → ing?)
- Lines 843-845: $𝒜$distinct → $𝒦_{\mathrm{distinct}}$, $𝒜$duplicate → $𝒦_{\mathrm{duplicate}}$, $𝒜_{\mathrm{dummy}}$ → $𝒦_{\mathrm{dummy}}$
- Lines 856-859: The four instances of $\mathcal{A}$ should all be replaced with $\mathcal{K}$.
- Line 954: $r_1=\delta^{-1}-1$ → $r_1=\delta^{-1}$
- Line 1027: affine (linear) maps
- Lines 1041-1043: The four instances of $\frac{k}{\delta}$ should be $k\delta$.
- Lines 1048, 1051: $k\delta+1$ → $(k+1)\delta$
- Line 1051: $k=0,1,\ldots,\delta^{-1}-1$ → $k\in\{0,1,\ldots,\delta^{-1}-1\}$
- Lines 1060-1062: The four instances of $\frac{k^2}{\epsilon}$ should be multiplied with $\delta$.
- Lines 1087-1093: The second column of the matrix in the argument of $\sigma_R$ should be multiplied with $\delta$.
- Line 1113: $\xi_1$ → $\xi_0$
- Line 1119: $d$-times (more) parameters.
- Line 1133: the self-attention layer select(s)
- Lines 1135, 1204: Better phrased as "in such a way that if it is selected, then it is replaced with a negative value ..."
- Line 1156: (is → are) ordered
- Line 1165: denote(s)
- Line 1170: $\delta^{-id}$ → $\delta^{-(k-i)d}$
- Line 1199: $\delta^{-1}-1$ → $\delta^{-d}-1$
- Line 1201: I think that the conditions imposed on $\epsilon$ and $M$ are incorrect. They should probably be $0\lt\epsilon\lt1$ and $M\gt\frac{\delta^{-d}-1}{\delta^{-d}-1+\epsilon}$. The same should also apply to the proof of Corollary A.6. For $x_1\in\{0,1,\ldots,(\delta^{-1} → \delta^{-1}-1)\}$
- Lines 1223, 1226: $\delta^{-1}$ → $\delta^{-d}$

---

### Decision · Program_Chairs · 2025-01-22

Reject